# The applicability and challenges of black carbon sensors in monitoring networks

J. Tapio Elomaa[1], Krista Luoma[1,2], Sami D. Harni[2], Aki Virkkula[2], Hilkka Timonen[2], and Tuukka Petäjä[1]

[1]Institute for Atmospheric and Earth System Research / Physics, Faculty of Science, University of Helsinki, Helsinki, Finland
[2]Atmospheric Composition Research, Finnish Meteorological Institute, Helsinki, Finland

*Correspondence to*: J. Tapio Elomaa (tapio.elomaa@helsinki.fi)

**Abstract.** Black Carbon (BC) is a particulate pollutant emitted as a by-product of combustion. BC has an emerging role in
air quality monitoring with the current recommendations by the World Health Organization to monitor BC to capture its temporal and spatial variability. To observe this variability, especially in urban areas, a large quantity of sensor-type measurements is required. In this study, four different types of small-scale filter-based BC sensors (AE51, MA200, MA350, and Observair) were used to build a sensor network in Kumpula campus, Helsinki, Finland. Our aim was to test the applicability of the sensors to monitor ambient BC concentrations in field conditions and to study the variation of BC at high
resolution. The results were compared to a reference level instrument (MAAP) for validation. During intercomparisons, the sensors had a good correlation with the reference and, after a simple orthogonal regression calibration, were deemed suitable for deployment in the sensor network. During deployment, the sensor network proved to be able to capture small-scale temporal and spatial differences in BC concentrations. Changes in temperature ($T$) and relative humidity ($RH$) were observed to induce error in the BC measurements. This error was amplified by the dualspot correction, which was worsening the
measurement result under unstable conditions of $T$ and $RH$. This should be considered when using sensors that apply the dualspot correction automatically. The environmental compensation used by the Observair sensors reduced the error from the changing $T$ and $RH$. To reduce the effect of changing $T$ and $RH$, more robust environmentally controlled boxes should be developed, or correction algorithms, such as environmental compensation, should be applied.

## 1 Introduction

Black carbon (BC) is a typical aerosol particle component in the atmosphere. BC consists of carbonaceous material that efficiently absorbs light at visible wavelengths and therefore appears black. It is emitted into the atmosphere as a by-product of incomplete combustion, such as traffic and biomass combustion. BC has remarkable effects on both climate and air quality (Bond et al., 2013).

BC affects the climate directly by interacting with solar radiation and indirectly via complex aerosol-cloud interactions (Stocker et al., 2013). Due to its absorbing nature, BC has a warming effect on the climate. The warming effect is enhanced if BC is emitted or transported in polar areas, where it speeds up the melting of snow and ice sheets by deposition (Sand et al., 2013; Kang et al., 2020; Räisänen et al., 2022).

From the air quality viewpoint, BC is an air pollutant with adverse health effects. Since BC particles fall typically in the size range of ultrafine particles (diameter < 100 nm), they can be transported into the deepest part of the human respiratory system, from there to the blood circulation system, and eventually end up even in the brain and other vital organs (Janssen et al., 2011; Segersson et al., 2017). Combustion-related emissions consist of large concentrations of other fine particles and toxic materials that have been shown to have more adverse health effects than particulate matter from other sources (Krzyzanowski et al., 2005). BC, as a by-product of combustion, has been shown to be a better indicator of the adverse health effects of atmospheric aerosol particles than the more commonly monitored mass of particles smaller than 2.5 µm in diameter ($PM_{2.5}$) (Janssen et al., 2011). In the long run, inhaled fine aerosol particles can cause cardiovascular and respiratory diseases as well as cancer (Ravindra, 2019; Lequy et al., 2021). Lelieveld et al. (2015) estimated that globally, exposure to $PM_{2.5}$ causes 1.9 million premature deaths per year.

In the recent air quality guidelines, WHO recommends starting systematic measurements of BC in urban areas to reduce the uncertainty related to the temporal and spatial variability of BC concentrations as well as its health, air quality, and climate impacts (WHO, 2021). Even though the recommendations to monitor BC, there are yet no limit values regarding BC concentration due to lack of epidemiological exposure studies.

Especially in urban areas, the concentration of BC can vary depending on both anthropogenic and natural factors: e.g., changing traffic rate, local biomass combustion, and weather conditions, orography, or close by buildings that affect the dilution by wind or convection (Helin et al., 2018; Caubel et al., 2019; Luoma et al., 2021b). For example, BC concentrations are halved by moving 30 m away from a busy traffic lane (Enroth et al., 2016). Due to these various sources and rather short lifetime (days compared to years with greenhouse gases), BC has a lot of temporal and spatial variation within urban districts and communities (Patrón et al., 2017; Caubel et al., 2019; Luoma et al., 2021b).

To capture and measure the spatial and temporal variability of BC in urban areas, one option is to deploy a high-resolution sensor network (Caubel et al., 2019). This requires a large quantity of affordable but robust sensors that can be deployed outside in ambient conditions. A viable option is to utilize commonly used filter-based methods that are robust, easy-to-use, and have a high time resolution. In the last decade small-scale versions of the filter-based instruments have been introduced reducing the cost of the sensors in relation to large monitoring instruments by sacrificing some reliability, sensor lifetime and accuracy (Kamboures et al., 2013; Caubel et al., 2018; Holder et al., 2018). Previous studies have reported a good correlation between BC sensors and reference-grade instruments, but with varying slopes and intercepts depending on location and sensor implicating the need for onsite calibration (Alas et al., 2020; Kuula et al., 2020; Chakraborty et al., 2023; Wu et al.,

2024). In previous studies, a common application for these sensors has been personal BC exposure as a carry-on measurement device (Delgado-Saborit, 2012; Li et al., 2015), mobile measurements (Alas et al., 2019; Pikridas et al., 2019), and sensor networks (Caubel et al., 2019).

The large quantity of sensors inevitably causes technical challenges, for example with maintenance, data acquisition, survivability of the sensors under the changing ambient conditions such as diurnal temperature changes and rain, sensor to sensor variability and internal sensor drift (Petäjä et al., 2021; Zaidan et al., 2023). Before a wide implementation of sensor networks, pilot deployments are needed to identify the challenges of individual sensor operations and sensor networks. Operating a variety of sensors side-by-side in the same network allows assessment of performance characteristics of different models of BC sensors, and to identify the critical qualities of a good small-scale BC sensor.

The aim of this study is to explore the suitability of four distinct types of filter-based small-scale BC sensors (AE51, MA200, MA350, Observair) for mapping the spatio-temporal variation of urban BC concentrations. To ensure the measurement quality, we compared the sensors with a Multi-Angle Absorption Photometer (MAAP) (Petzold and Schönlinner, 2004) in two intercomparison periods at Station for Measuring Ecosystem–Atmosphere Relations III (SMEAR III, (Järvi et al., 2009)) in Kumpula campus, Helsinki, Southern Finland, from the end of May to start of October 2022. In between the two intercomparisons, the sensors were deployed as a sensor network in the surrounding Kumpula campus area. We characterized the applicability of the different sensor types within the sensor network, and the suitability and challenges regarding their utilization in ambient measurements. Furthermore, we provide preliminary results for the general features of BC concentrations within the Kumpula campus area and its spatio-temporal variation.

## 2 Methods

### 2.1 Measuring principle to obtain BC mass concentration with small scale sensors

Filter-based optical methods are widely used to measure BC concentration due to their ease of operation and relatively low cost (Hansen et al., 1984). With this technique, sample air is drawn through a filter material, where aerosol particles are collected onto the filter. The attenuation of light through the filter area increases over time due to increased absorption and scattering from the collected particles. The attenuation is described by Eq. 1, where $I_0$ is the light intensity through a clean filter and $I$ is the light intensity through a loaded filter:

$$ATN = -\ln(I/I_0)$$

( 1 )

The attenuation coefficient $b_{ATN}(\lambda)$ [m⁻¹] is calculated from the measured light intensity and the operational parameters of the instrument as described in Eq. 2, where $A$ [m²] is the area of the sample spot, $Q$ [m³ s⁻¹] is the volumetric flow through the sample spot, $\Delta t$ [s] is the collection time and $\lambda$ is the wavelength of the light source.

$$b_{ATN}(\lambda) = \frac{A}{Q}\frac{\Delta ATN(\lambda)}{\Delta t} \qquad (2)$$

To determine the BC concentration from the attenuation coefficient, a series of assumptions are necessary, and some corrections need to be applied. The attenuation consists of: (1) absorption from the aerosol particles, (2) enhanced attenuation from multiple scattering by the filter fibers (multiple scattering), (3) enhanced attenuation from scattering of the aerosol particles (aerosol scattering) and (4) the saturation of the filter which causes the attenuation to change non-linearly over time (loading effect) (Collaud Coen et al., 2010). In a general form the BC calculation can be presented as

$$eBC = \frac{1}{MAC(\lambda)} \cdot \sigma_{ap}(\lambda) = \frac{1}{MAC(\lambda)} \cdot \frac{f(ATN)b_{ATN}(\lambda) - s(\lambda)\sigma_{sp}(\lambda)}{C_{ref}} \qquad (3)$$

where $\sigma_{ap}(\lambda)$ [m$^{-1}$] is the absorption coefficient (1), $C_{ref}$ is the multiple scattering correction factor (2), $s(\lambda)$ is a fraction of the scattering coefficient $\sigma_{sp}(\lambda)$ [m$^{-1}$] (3), $f(ATN)$ is a loading correction function (4), and $MAC(\lambda)$ [m$^2$ g$^{-1}$] is the mass absorption cross section (*MAC*) (Virkkula et al., 2015). The results are given as equivalent black carbon (*eBC*) denoting the conversion of the absorption coefficient to mass concentration with the use of a specific *MAC* value (Petzold et al., 2013).

It is assumed that with an 880 nm light source the absorption is only from BC particles minimizing the effect of absorbing organic carbon species (i.e., brown carbon, BrC), which absorb light only on shorter wavelengths. Hence in this study, the *eBC* concentration is determined at $\lambda = 880$ nm (apart from MAAP that operates at 637 nm). The multiple scattering factor $C_{ref}$ depends on the filter material and instrument used. Most commonly a constant value is used appropriate for the instrument and filter material. It is to be noted that the $C_{ref}$ value can have a large variability depending on seasons, location, and methodology of determination (Collaud Coen et al., 2010; Backman et al., 2017; Di Biagio et al., 2017; Bernardoni et al., 2021; Luoma et al., 2021a). The aerosol scattering correction requires measurement of the $\sigma_{sp}$, which in many cases is not possible due to the lack of instrumentation. Due to this the aerosol scattering correction is often disregarded as in this study. For the loading correction, a plethora of options are available (Bond et al., 1999; Weingartner et al., 2003; Arnott et al., 2005; Schmid et al., 2006; Kirchstetter and Novakov, 2007; Virkkula et al., 2007; Collaud Coen et al., 2010; Hyvärinen et al., 2013; Drinovec et al., 2015; Chakraborty et al., 2023). In this study, the dualspot correction (Drinovec et al., 2015; Chakraborty et al., 2023) was tested for the sensors. The correction was selected as it is the most recent one, it is widely used with Aethalometer® model AE33 and capability of this correction is inbuilt to the design of MA200 and MA350 sensors that were utilized in this campaign (see Sect. 2.2). For the *MAC* value a constant value is commonly used with the assumption that the measured BC is freshly emitted (Bond and Bergstrom, 2006; Bond et al., 2013; Liu et al., 2020).

## 2.2 Dualspot correction algorithms

The dualspot correction is a scheme to correct for the loading effect by relating two measurement spots with differing flows.
The correction is presented in Eq. 4, where $eBC_{\text{NC}}$ is the uncorrected measurement and $k$ is the compensation parameter.

$$eBC = \frac{eBC_{\text{NC}}}{(1 - k \cdot ATN)} \qquad (4)$$

The $k$ can be determined numerically from the overall loading of the two filter spots as presented in Eq. 5, where subindices $L$ and $H$ refer to the low and high flow spots, respectively (Drinovec et al., 2015). $FVRF$ is the face velocity ratio factor.

$$\frac{Q_{\text{L}}}{Q_{\text{H}}} \cdot FVRF = \frac{ln(1 - k \cdot ATN_{\text{L}})}{ln(1 - k \cdot ATN_{\text{H}})} \qquad (5)$$

As the $k$ is very sensitive to errors in sample flow measurements, the additional empirical factor $FVRF$ is implemented to reduce the sample flow measurement uncertainty. The $FVRF$ is calculated by plotting $ATN_{\text{L}}/ATN_{\text{H}}$ to $ATN_{\text{H}}$ and taking the intercept of a linear fit. The linear fit is done when $ATN_{\text{H}}$ is between $ATN_{f1}$ and $ATN_{f2}$ with example values being 10 and 30 respectively. The lower limit ($ATN_{f1}$) is set to minimize the effect of particle transients in the fresh filter spot and the upper limit ($ATN_{f2}$) is set low enough so that the data are not yet affected by the loading effect. This should ensure that at the low loading the $ATN$ and flow ratios of the two spots are proportional to each other and therefore the sample flow measurement error can be minimized with the $ATN$ measurements.

Due to the determination of the $FVRF$ and the $k$ being unstable at low loadings and more accurate at high loadings, the $k$ is additionally weighted according to Eq. 6, where $k_{\text{w}}$ is the weighted $k$, $ATN_{\text{TA}}$ is the tape advance trigger (default $ATN_{\text{TA}} = 120$ at 370 nm) and $k_{\text{old}}$ is the $k$ calculated from a previous filter spot i.e. before the tape advance:

$$k_{\text{w}} = \frac{(ATN_{\text{TA}} - ATN_{\text{H}})k_{\text{old}} + (ATN_{\text{H}} - ATN_{f2})k}{(ATN_{\text{TA}} - ATN_{f2})} \qquad (6)$$

All in all, the weighing results in that for most of the time the static $k_{\text{old}}$ value is used, and the real time determined $k$ according to Eq. 5 rises in importance at higher loadings and closer to the times when the tape advance is triggered. The final weighted $k_{\text{w}}$ at full loading is equal to the real time determined $k$. The correction is applied to the high flow spot with the weighted $k_{\text{w}}$ and Eq. 4 real time during the measurements.

The MA200 and MA350 sensors utilize a variation of the dualspot correction (Chakraborty et al., 2023; Mendoza et al., 2024). In this version the k is calculated as seen in Eq. 7 and no weighting is used. The data is corrected with the $k_{MA}$ as seen in Eq. 4.

$$k_{MA} = \frac{eBC_L - eBC_H}{(eBC_L * ATN_H) - (eBC_H * ATN_L)} \quad (7)$$

In this study both versions of the correction were tested.

### 2.3 Deployment of small BC sensors at the Kumpula Campus

We used four types of small-scale black carbon sensors and one reference instrument. The sensors were AE51 (2 units), MA200 (1 unit) and MA350 (1 unit) by Aethlabs and Observair (OBS, 4 units) by Distributed Sensing Technologies (DST). As the reference instrument, we used the Multi-Angle Absorption Photometer (MAAP) by Thermo Fischer Scientific
(Petzold and Schönlinner, 2004).

The sensor specifications are given in Table 1. All small-scale sensors can be operated with flow rates between 50–200 ml min$^{-1}$. AE51 and Observair measure with one wavelength at 880 nm while the MA-sensors measure with 5 wavelengths (880, 625, 528, 470, and 375 nm). All sensors calculate the *eBC* concentration at 880 nm according to the assumption to minimize the effect of *BrC*. The other wavelengths of the MA-sensors can be used to differentiate between *BrC* and *BC* and
the possible sources of these particles. In this study only the 880 nm wavelength was utilized to conform to the other sensors. The MA-sensors have an inbuilt capability for the dualspot correction and therefore they have two separate measurement spots and one reference spot. The AE51 and Observair have one measurement spot and one reference spot. The AE51 and Observair sensors were run in pairs for the prospect of the dualspot correction (see Sect. 3.2.). The filter materials were for the AE51 Teflon coated quartz fibers (T60), for the MA- sensors polytetrafluoroethylene (L15 or L85 filter cartridge), and
for the Observair the filter material was described as fibrous filter material (Distributed Sensing Technologies, 2023). AE51 and Observair had single-use filters that needed to be replaced regularly, in our case every 4–5 days. The MA-sensors have filter cassettes that automatically change the filter spot after a high loading limit is reached. In our case the filter was set to change when *ATN* was higher than 100 at any wavelength (most likely the lowest wavelength of 375 nm), but the setting can be changed between 1–100. The MA200 filter cassette has 15 spots and the MA350 has 85 spots. The filter sample spot is
7.1 mm$^2$ for all sensor types and all sensors use the same *MAC* = 7.8 m$^2$ g$^{-1}$ (at 880 nm). The $C_{ref}$ value for AE51 is 1.6 and for the other sensors 1.3. All sensors have additional measurements of temperature (*T*) and relative humidity (*RH*) and the Observair sensors utilize environmental compensation technology to compensate for sharp changes in *T* or *RH*.

**Table 1**. *The technical details of the black carbon sensors and the multi-angle absorption photometer (MAAP) used in this study.*

| Parameter | AE51 | MA200 | MA350 | Observair | MAAP |
|---|---|---|---|---|---|
| Flow rate [ml min$^{-1}$] | 50–200 | 50–150 | 50–150 | 50–200 | 5000 |
| Number of wavelengths | 1 | 5 | 5 | 1 | 1 |
| Measurement interval [s] | 1–300 | 1–300 | 1–300 | 2–60 | 300 |
| Filter material | Teflon coated quartz fiber | Polytetrafluoroethylene | Polytetrafluoroethylene | Fibrous | Glass fiber (GF10) |
| Filter usage | Single | A cassette with 15 spots | A cassette with 85 spots | Single | Tape ~40 m |
| Sample spot area [mm$^2$] | 7.1 | 7.1 | 7.1 | 7.1 | 200 |
| $MAC$ [m$^2$ g$^{-1}$] (at 880 nm) | 7.8 | 7.8 | 7.8 | 7.8 | 6.6 (at 637 nm) |
| $C_{ref}$ | 1.6 | 1.3 | 1.3 | 1.3 | Measured |
| Limit of detection [ng m$^{-3}$] | $\pm100$ (1 min avg, 150 ml min$^{-1}$) | $\pm30$ (5 min avg, 150 ml min$^{-1}$) | $\pm30$ (5 min avg, 150 ml min$^{-1}$) | $\pm50$ (not specified) | $\pm50$ (20 min, 16.67 l min$^{-1}$) |

The reference instrument MAAP is also a filter-based absorption photometer, but it differs from the measurement principle presented in Sect. 2.1 by additionally measuring backscattering from the filter at two angles to improve the accuracy of the $\sigma_{ap}$ and $eBC$. Additionally, the MAAP derives the $\sigma_{ap}$ by applying a two-stream-approximation radiative transfer scheme (Petzold and Schönlinner, 2004). Therefore, it is somewhat more independent measurement method and is a good reference instrument for the $eBC$ sensors. MAAP has been used as a reference instrument also in previous studies comparing filter-based instruments (e.g. Alas et al., 2019; Luoma et al., 2021a). The reported uncertainty and unit-to-unit variability of MAAP (at 16.67 l min$^{-1}$ flow) are 12% and 3% (Petzold and Schönlinner, 2004; Müller et al., 2011). Here, the flow rate was

set to 5 l min$^{-1}$. The instrument measures with only one wavelength at 637 nm (Müller et al., 2011) and applies MAC = 6.6 m$^2$ g$^{-1}$ (at 637 nm). The filter tape is made with glass fiber and the tape advance is automatic. In our case, the filter tape needs to be changed on average every 6 months. The measurement spot is considerably larger, 2 cm$^2$ in comparison to the small-scale sensors, which was 7.1 mm$^2$.

## 2.4 Description of the sampling site

The field campaign was conducted at the Kumpula campus located approximately 4 km northeast from the center of Helsinki, Finland. Helsinki is the capital of Finland located in the south at the coast of the Gulf of Finland. The metropolitan area consists of four cities with a combined population of 1.2 million people (Statistics Finland: Population, Date accessed 01/12/2023, 2023). The main sources of BC in the region are from road traffic, wood burning, maritime traffic and transboundary air pollution (Helin et al., 2018; Teinilä et al., 2022). In 2022, the air quality in the region was good or satisfactory 90% of the time (Helin et al., 2018; Korhonen et al., 2022; Teinilä et al., 2022).

The Kumpula campus was selected as the study area due to easy access for deployment, maintenance, and upkeep. The surrounding area consists of parks, detached housing zones and a relatively high-capacity road (Järvi et al., 2009). In addition, there was an active construction site in the area during the measurements (Fig. 1). Two intercomparison periods were measured during 26.5. – 6.6.2022 (11 d) and 16.9. – 3.10.2022 (17 d) at the Station for Measuring Ecosystem–Atmosphere Relationships III (SMEAR III, 60°12′N, 24°58′E, 26 m above sea level) (Järvi et al., 2009). In between the intercomparisons 4.7. – 16.9.2022 (74 d), the sensors were deployed to the locations seen in Fig. 1.

Kumpula campus is located on a small hill 26 m above sea level and the area with the surroundings is presented in Fig. 1. Southwest from the Kumpula campus is the Kumpula botanical garden and park area with trees and vegetation. In the center of the campus lies the university buildings, Finnish meteorological institute (FMI), few four-story apartment blocks, and a construction site. Further north there is a low-density residential area of mainly wooden houses with more park area. On the eastern side, there is a road to the city center, Kustaa Vaasan tie, which is used by approximately 38000–42000 vehicles per day with around 10% being heavy vehicles (Helsinki city road statistics, Date accessed: 01/02/2024, 2024). Beyond the road lies Toukola residential area with much larger apartment blocks in comparison to the northern side and a small shopping center. The campus area has a bus line going through it with the bus stops marked as small blue squares in Fig 1. Locally, BC is emitted by traffic and wood combustion on the detached housing areas and communal garden.

During the intercomparisons at the SMEAR III, the reference instrument MAAP was used with a pre-impactor removing particles larger than 1 µm in diameter from the sample flow. The inlet was positioned about 7 m height from the ground. The small-scale sensors were all measuring on the same sample line (different with MAAP) that did not have any inlet pre-impactor. The separate measurement line was set up through the SMEAR III station wall at a height of 3 m from the ground.

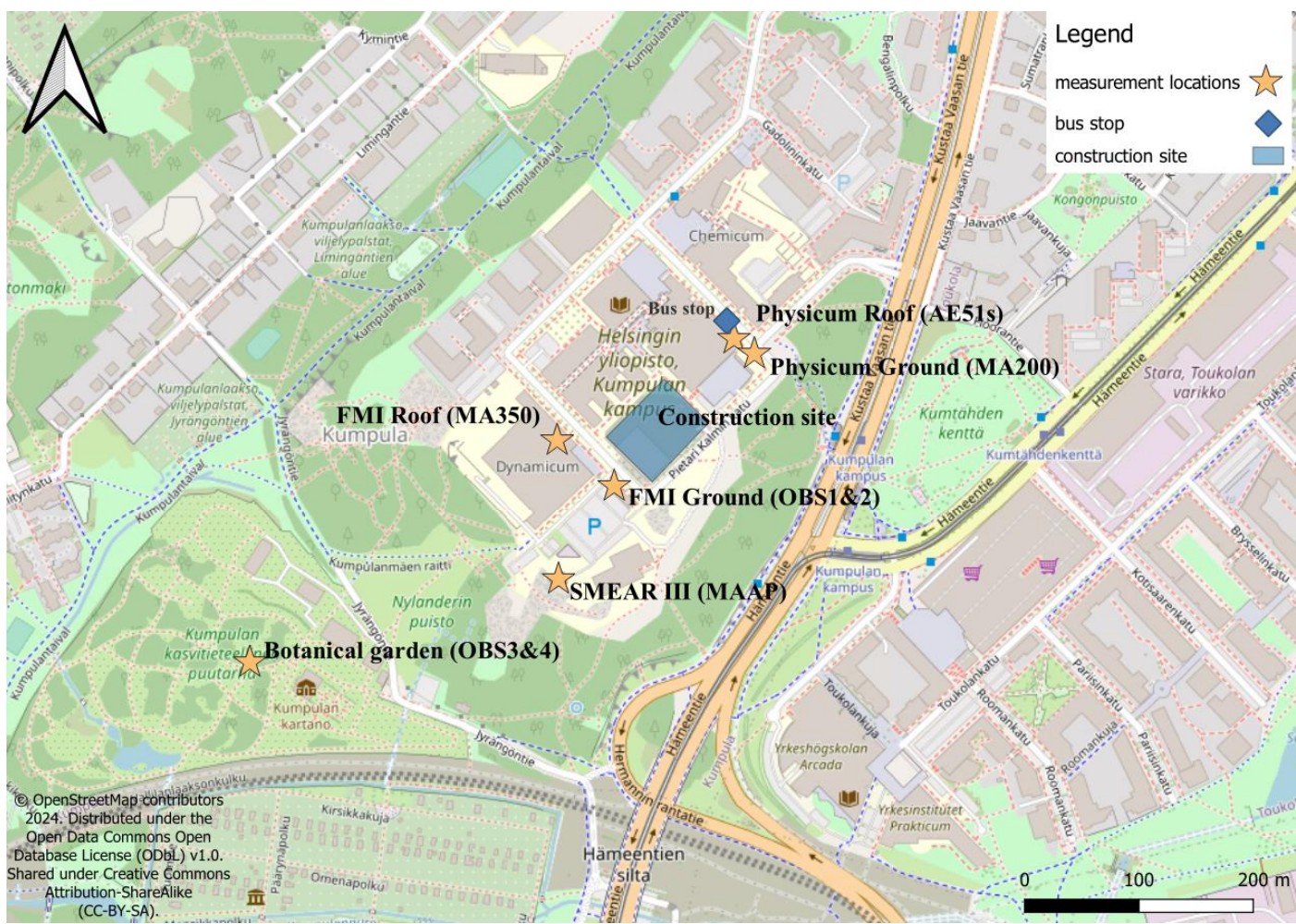

**Figure 1**. *Map of the deployment locations in the Kumpula campus and the surrounding area. Also, the construction site and the bus stop close to "Physicum Roof" measurement are marked on the map. Shared under Creative Commons Attribution-ShareAlike (CC-BY-SA).*

The deployment locations are described in Table 2. In some locations, two sensors were deployed for redundancy and the possibility of applying the dualspot correction manually. For the dualspot the pairings were run with differing flow rates. The flow rates used during the different phases of the campaign are outlined in Table 3. The closest sources to the locations were a bus stop near the Physicum roof ($P_{roof}$) and Physicum ground ($P_{ground}$) locations on a small road. The FMI parking lot in the middle of the FMI roof ($F_{roof}$), FMI ground ($F_{ground}$) and the $SMEARIII_{ground}$ locations. The last location, Kumpula botanical garden ($BG_{ground}$) has minimal traffic. The sensors were flow calibrated during the measurement campaign and the timing and results are outlined in the supplement.

**Table 2**. *Information of eBC sensor deployment locations (height, type, container), deployment duration (Full: 4.7. – 16.9.2022 (74 d), partial: 4.7.–19.7.2022 (15 d)), and indicated issues related to their operation during the deployment.*

| Location | $P_{roof}$ | $P_{ground}$ | $F_{roof}$ | $F_{ground}$ | $BG_{ground}$ | $SMEARIII_{ground}$ |
|---|---|---|---|---|---|---|
| Height | 15 m | 1.2 m | 18 m | 1.2 m | 1.2 m | 7 m |
| Sensor type and ID | $AE51_{1408}$ $AE51_{1409}$ | MA200–0187 | MA350–0104 | OBS1 (OBS_15) OBS2 (OBS_71) | OBS3 (OBS_74) OBS4 (OBS_37) | MAAP |
| Container | Indoors | B&W Type 3000 | B&W Type 3000 | Observair's own box | B&W Type 3000 | Indoors |
| Deployment duration | Full | Full | Partial | Full | Full | Full |
| Issues | No issues | Temperature dips | Overheating, inlet blocked since 22.7., full failure of spot 2 on 19.7. | Overheating and low battery | Occasionally overheating, missing 11.–17.8. OBS3, 13.–21.7. OBS4, due to low battery | No issues |
| Notes | Bus stop | Bus stop | Breakdown during deployment | Parking lot | Minimal car traffic in the area | Parking lot |

**Table 3.** *Flow rates used during the measurements. Not Available (N.A.) is listed for the sensor data sets, which were not available due to instrument failures.*

| | 1st Intercomparison | | Deployment | 2nd Intercomparison |
|---|---|---|---|---|
| Sensor | 26.5.–31.5.2022 (5d) | 1.6.–6.6.2022 (6d) | 4.7.–16.9.2022 (74 d) | 16.9.–3.10. 2022 (17 d) |
| $AE51_{1408}$ | 150 ml min$^{-1}$ | 100 ml min$^{-1}$ | 100 ml min$^{-1}$ | 100 ml min$^{-1}$ |
| $AE51_{1409}$ | 150 ml min$^{-1}$ | 200 ml min$^{-1}$ | 200 ml min$^{-1}$ | 200 ml min$^{-1}$ |
| MA200 | 150 ml min$^{-1}$ | | 150 ml min$^{-1}$ | 150 ml min$^{-1}$ |
| MA350 | 150 ml min$^{-1}$ | | 150 ml min$^{-1}$ | N.A. |
| OBS1 | 145 ml min$^{-1}$ | | 135 ml min$^{-1}$ | N.A. |
| OBS2 | 100 ml min$^{-1}$ | | 100 ml min$^{-1}$ | N.A. |
| OBS3 | 145 ml min$^{-1}$ | | 145 ml min$^{-1}$ | 145 ml min$^{-1}$ |
| OBS4 | 100 ml min$^{-1}$ | | 100 ml min$^{-1}$ | 100 ml min$^{-1}$ |

## 2.5 Data analysis

During data processing, data were removed near filter changes. The filter changes were manually identified, and two hours of data were removed starting from the nearest hour before the filter change. This was done for all small-scale sensors.

During the deployment starting from 19.7.2022 the MA350 at $F_{roof}$ had flows significantly lower than the set value. This was most likely due to inlet blockage and the start of a pump failure. Data were removed from this point forward as it was deemed erroneous. The sensor suffered a total pump failure after it was moved to SMEAR III for the 2nd intercomparison (see Sect. 3.2.4).

OBS3&4 located at $BG_{ground}$ had shutdowns due to low battery during the deployment. After the sensor restart the data had erroneous starting spikes. Two hours of data were removed starting from the nearest hour before the restarts. Due to the shutdowns a missing section of OBS3 data are patched with OBS4 data during the deployment. This was done so that the $BG_{ground}$ location has a continuous time series. The sensor-to-sensor variability was deemed low enough as a justification for this process.

In total between 1.5–2.9 % of the available data was removed for all sensors except MA350 for which 69.1 % of the data was removed that was most of the deployment period. Note that OBS1, OBS2 and MA350 were not tested in intercomparison 2 due to breakage.

For calibration an *F*-factor was calculated as seen in Eq. 8 using data from the 1[st] intercomparison. The corresponding sensor data was then multiplied by the reciprocal of this value.

$$F = \frac{eBC_{\text{sensor,mean}}}{eBC_{\text{MAAP,mean}}}$$
(8)

Python3 was used for most of the data analysis with numpy, scipy, matplotlib, pandas, seaborn and mpl-scatter-density packages (Harris et al., 2020; Hunter, 2007; pandas development team, 2020; Virtanen et al., 2020; Waskom, 2021, https://github.com/astrofrog/mpl-scatter-density, accessed 21.8.2024). For the wind plots R with the openair library were used (Carslaw and Ropkins, 2012).

## 3. Results and discussion

### 3.1 Intercomparison of BC sensors

Before and after the deployment, intercomparison measurements were conducted at the SMEAR III to study the differences between the sensor types and the individual units. The sensors were measuring ambient *eBC* concentrations parallel with the reference instrument MAAP (see Sect. 2.4). The intercomparison measurements were conducted during 26.5.2022–6.6.2022 (11 d) and 16.9.2022–3.10.2022 (17 d). All the sensors were tested in the 1[st] intercomparison (AE51 x2, MA200, MA350, OBS x4). During the deployment MA350, OBS1 and OBS2 were damaged and therefore were not tested in the 2[nd] intercomparison. Time series of the intercomparisons can be seen in Fig. 2 in 5-min averages. Correlation of all the sensors in relation to the reference instrument MAAP is seen in Fig. 3 with an orthogonal regression. The values of the orthogonal regression line fit are listed in Table 4. For MA-sensors, spot 1 data are used instead of dualspot corrected data (see Sect. 3.1.2).

With 5-min averaging all sensors showed a good Pearson correlation (*r*) between 0.78–0.85 during the 1[st] intercomparison period. Results of AE51 sensors were very comparable with both having an intercept of 42 and slope of 0.84. During this time AE51$_{1408}$ and AE51$_{1409}$ were run with a flow rate of 150 ml min$^{-1}$ between 26.5.2022–31.5.2022 (5 d) and 100 ml min$^{-1}$ and 200 ml min$^{-1}$ between 1.6.2022–6.6.2022 (6 d) respectively (Table 3). During the second intercomparison, there was a larger difference, where the AE51$_{1409}$ had a weaker intercept, slope, and *r* of 55.6, 0.70 and 0.92 in comparison to the

respective values for AE51$_{1408}$ of 48.5, 0.78 and 0.94. Both sensors showed improved $r$ but weaker slope and intercept. Alas et al. (2020) has reported similar results in different types of environments with AE51 compared to MAAP with reduced major axis (RMA) regression. In Manila during summer of 2015 AE51 had $r$ of 0.845 with a slope of 0.871 ± 0.013 and in Rome during winter of 2017 better $r$ of 0.983 and slope of 1.015 ± 0.003. In Loški Potok, with AE33 as the reference, the reported intercomparison values for rural background were r = 0.962 with slope = 0.876 ± 0.005 and for rural village $r$ = 0.978 with slope = 0.826 ± 0.002. Varying slopes are most likely caused by different aerosol types that depend on the location and season.

The MA-series sensors showed similar results where the sensors were comparable to each other with MA200 having intercept, slope, and $r$ of 51.5, 1.08 and 0.85 during the 1st intercomparison. The respective values for MA350 were 42.5, 1.13 and 0.83. MA350 did not survive for the 2nd intercomparison. The MA200 showed better performance during the 2nd intercomparison with $r$ = 0.92 and lower intercept of 28.6. The slope reduced to 0.90. The correlations of the MA-series sensors were comparable to the AE51 sensors, but on average the MA-series sensors measured slightly higher concentrations of $eBC$. Kuula et al. (2020) reported intercept, slope and $r$ of -44, 0.85 and 0.97 of MA350 when compared to AE33. Chakraborty et al. (2023) reported $r$ ~ 0.90 and slopes ranging from 0.736-1.01 for three MA300 units compared to AE33. As the AE33 measures slightly higher concentrations than MAAP (Pikridas et al., 2019; Wu et al., 2024) the intercomparison results seem to be inline with previous studies.

From the Observair sensors OBS1 was an older sensor that had been utilized in previous campaigns while OBS 2, 3 and 4 were new. The sensors showed very good comparability with $r$ in the range of 0.82–0.84 during the 1st intercomparison. The higher flow sensors (OBS1&3) measured slightly higher concentrations than the lower flow sensors (OBS2&4) with slopes being [1.06, 1.02] in comparison to [0.95, 0.91]. During the 2nd intercomparison the same pattern was observed, where OBS3 measured slightly higher concentrations compared to OBS4 with slope of 0.77 compared to 0.72. The reduction of slopes was more drastic during 2nd intercomparison with OBS sensors than AE51s or MA200. The $r$ improved to 0.88-0.91. Previous studies have reported similar $r$ and slightly lower slopes with AE33 as reference. r = 0.904 with slope = 0.57 (Wu et al., 2024) and r = 0.89 with slope = 0.87 (Caubel et al., 2018). Wu et al. (2024) noted that the low slope could be partially explained due to high loading of the filters. In this study filters were changed regularly and the average concentrations were much lower that in Wu et al. (2024) (mean 230 ng m$^{-3}$, 1465 ng m$^{-3}$).

Surprisingly, all sensors that were available performed better during the 2nd intercomparison, which is likely due to higher $eBC$ level during the 2nd intercomparison. The 1st intercomparison has on average lower concentrations compared to the 2nd intercomparison. This is due to the difference in meteorological conditions and possible differences in traffic density during these periods. Overall, the correlations between the sensor types were comparable, but there were slight differences on the base $eBC$ level between the sensor types.

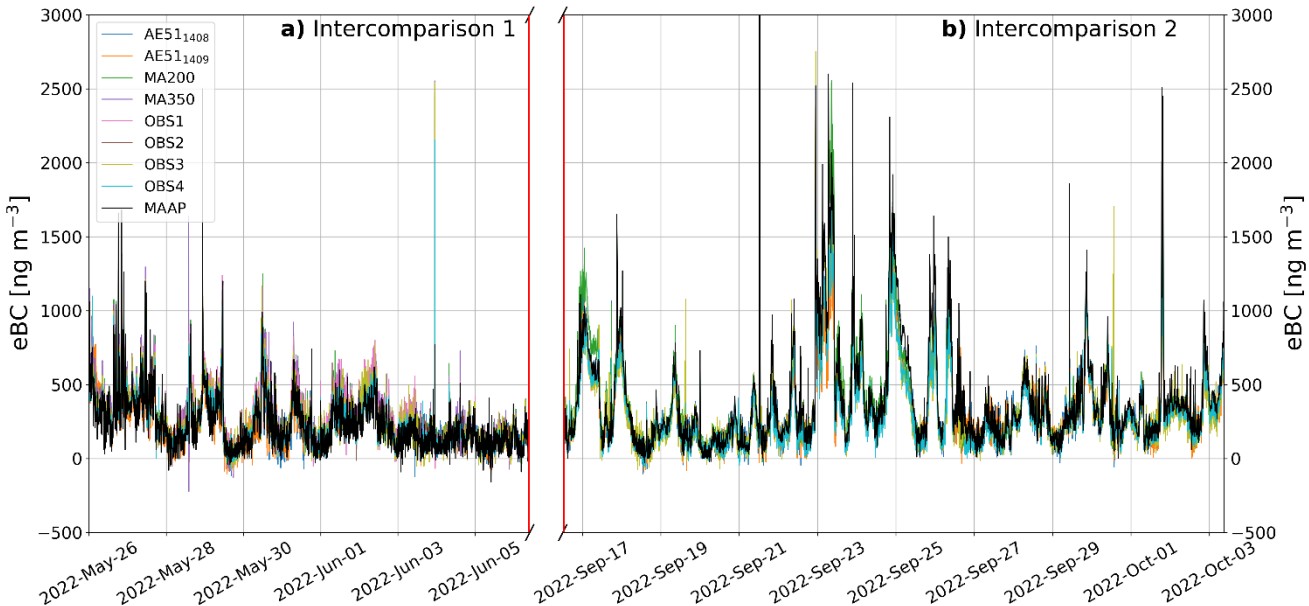

**Figure 2**. *Time series of both intercomparison periods a) 26.5.–6.6.2022 and b) 16.9.–3.10.2022. In the figure there is a split x-axis, where the period in between panels (a) and (b) marked with the vertical red lines is approximately 3.5 months. This period was the deployment phase between the intercomparisons. Data points are 5-min averages.*

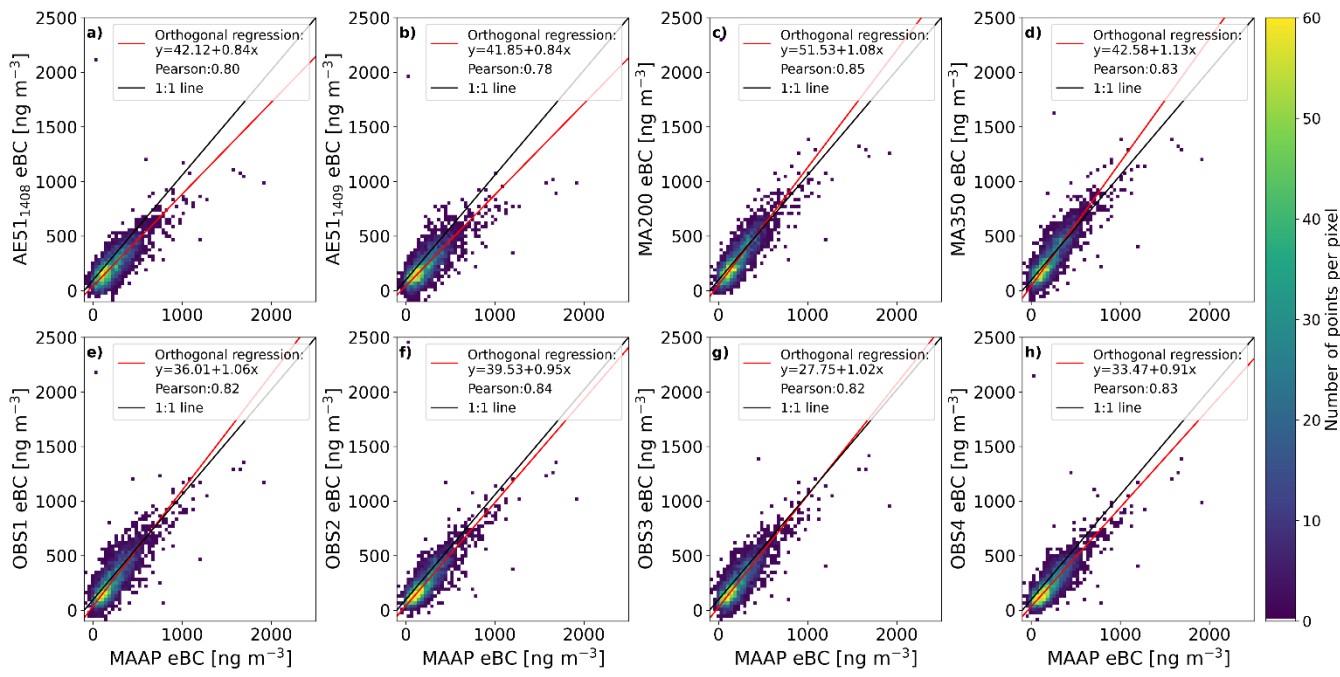

**Figure 3**. *Scatter density plot of the correlation between the eBC sensors and the reference instrument MAAP. Data are from 1st intercomparisons 26.5.–6.6.2022 as 5-min averages.*

**Table 4.** *Results of the intercomparison between the sensors and MAAP (5-min averages). Intercept and slope describe an orthogonal regression line fit (see Fig. 3 for the 1ˢᵗ intercomparison), and r is Pearson correlation coefficient. Not Available (N.A.) is listed for the sensor data sets, which were not available due to instrument failures.*

| | 1ˢᵗ intercomparison 26.5.–6.6.2022 | | | 2ⁿᵈ intercomparison 16.9.–3.10.2022 | | |
|---|---|---|---|---|---|---|
| Sensor | Intercept | Slope | $r$ | Intercept | Slope | $r$ |
| AE51$_{1408}$ | 42.1±2.81 | 0.84±0.01 | 0.80 | 48.5±2.15 | 0.78±0.004 | 0.94 |
| AE51$_{1409}$ | 41.9±3.01 | 0.84±0.01 | 0.78 | 55.6±2.18 | 0.70±0.004 | 0.92 |
| MA200 spot1 | 51.5±3.06 | 1.08±0.01 | 0.85 | 28.6±2.85 | 0.90±0.01 | 0.92 |
| MA350 spot1 | 42.6±3.43 | 1.13±0.01 | 0.83 | N.A. | N.A. | N.A. |
| OBS1 | 36.0±3.38 | 1.06±0.01 | 0.82 | N.A. | N.A. | N.A. |
| OBS2 | 39.5±2.87 | 0.95±0.01 | 0.84 | N.A. | N.A. | N.A. |
| OBS3 | 27.8±3.29 | 1.02±0.01 | 0.82 | 28.6±2.63 | 0.77±0.005 | 0.91 |
| OBS4 | 33.5±2.79 | 0.91±0.01 | 0.83 | 37.9±2.02 | 0.72±0.004 | 0.94 |

### 3.1.1 Applicability of the dualspot corrections

Dualspot corrections, that compensate for the loading effect, were tested during the intercomparison periods. The performance of the corrections can be seen in Fig. 4, where the sensor data and dualspot corrected data with both methods are compared to the reference instrument MAAP. For MA200 the $k_{\text{w}}$ version of the correction increased the difference to the reference from 21 to 132 ng m$^{-3}$ and with the $k_{\text{MA}}$ version from 21 to 48 ng m$^{-3}$. Most notably the variation of the differences increased in both cases, reducing the precision (seen as larger range of whiskers in Fig. 4) of the measurement. For MA350 the $k_{\text{w}}$ increased the difference from 67 to 145 ng m$^{-3}$ and the $k_{\text{MA}}$ decreased the difference from 67 to -22 ng m$^{-3}$. The precision was reduced, but not as much as for the MA200. For $k_{\text{MA}}$ the inverse in the compensation seems to arise from the relative differences of spot 1 and spot 2 and the calculation mechanism. The $k$ was observed to be highly variable and occasionally beyond reasonable values with both methods. The AE51 and Observair sensors were paired, and the corrections were applied manually by post-processing. For the AE51 the difference improved from -22 to 5 ng m$^{-3}$ with the $k_{\text{w}}$ method and to 8 ng m$^{-3}$ with the $k_{\text{MA}}$ method. The precision remained relatively constant with the $k_{\text{w}}$ method and decreased slightly with the $k_{\text{MA}}$ method. The correction worked by increasing concentrations at high *ATN* and increasing the accuracy of the measurement. For the Observair pairings the corrections increased the difference to the reference for OBS1&2 and for OBS3&4. The $k_{\text{w}}$ correction increased concentrations and the $k_{\text{MA}}$ decreased concentrations. For both pairings the corrections reduced the precision of the measurement.

Due to the reduction of the precision in most (4/5) cases, it was decided that the correction is not implemented during the deployment and spot 1 data are used with MA-series sensors. Instead, a simple calibration (see Sect. 3.1.2) was used to improve accuracy of the sensors in relation to the reference instrument MAAP. The use of dualspot correction was seen to be highly unstable with both correction methods.

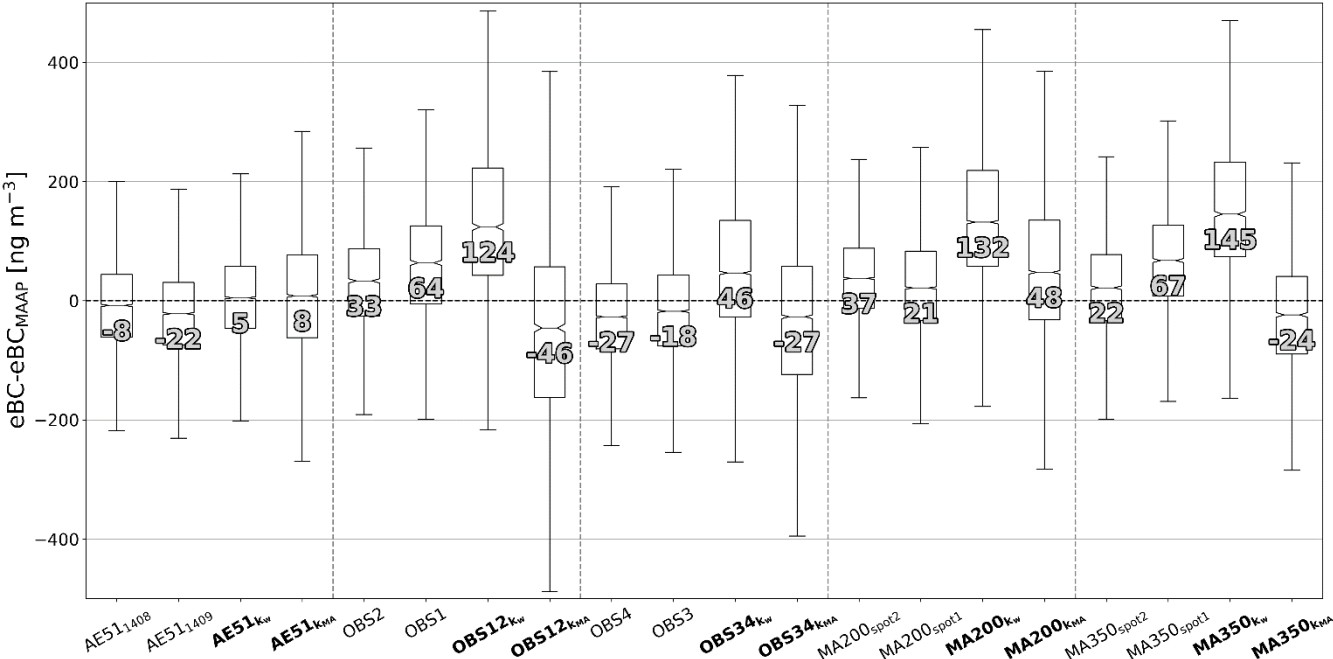

**Figure 4**. *The effect of the dualspot correction during the intercomparison periods. The dualspot correction is calculated with both correction factors k according to Eqs. 6 and 7. For the MA-sensors the k$_{MA}$ correction is calculated by the instrument. Data are from both intercomparison periods in 5-min averages. In the plot the middle line shows the median, top of box 75$^{th}$ percentile, bottom of the box 25$^{th}$ percentile and top and bottom whisker the last points within 1.5 times the interquartile range. The values are the medians of the corresponding boxes.*

### 3.1.2 Adjusting differences between sensors for comparison

To improve the accuracy and comparability of the sensor types, simple calibrations were applied to the data. Two calibrations were tested: *F*-factor and orthogonal regression line fit. The *F*-factor was calculated according to Eq. 8 and the orthogonal fit calibrations were calculated by applying the sensor respective equations as seen in Fig. 3 and Table 4 to the data. The results of the calibrations can be seen in Fig. 5.

The *F*-factor calibration reduced the spread of the data most aggressively. The medians agreed after the calibration within one standard error of the reference instrument. The orthogonal fit performed near equal to the *F*-factor calibration. For the MA-series the orthogonal calibration overcompensated slightly, but for the Observair sensors this method performed better. After the calibration mean and median values are within ±5 ng m$^{-3}$ for the Observair sensors, ±8 ng m$^{-3}$ for the MA-series and ±18 ng m$^{-3}$ for the AE51s. All sensor medians were within one standard error of the reference (MAAP) after calibration.

Fig. 6 and Table 5 shows the correlation between the data calibrated via the orthogonal fit and MAAP. The new orthogonal line fit intercepts and slopes are within ±4 ng m$^{-3}$ and ±0.05 respectively.

The orthogonal regression fit was selected as it considers variation of the sensors and the reference. The whole data set were calibrated according to the orthogonal fit equations determined from the 1$^{st}$ intercomparison. During the analysis this calibration step was observed to be imperative as it reduced the differences between the locations during the deployment phase. Without the calibration differences between locations could have been incorrectly seen as differences in sources, when in fact they were just differences between the instruments. Similar approach has been used before by Petäjä et al. (2021) for cost-effective gas and PM$_{2.5}$ analyzers for urban air quality measurements.

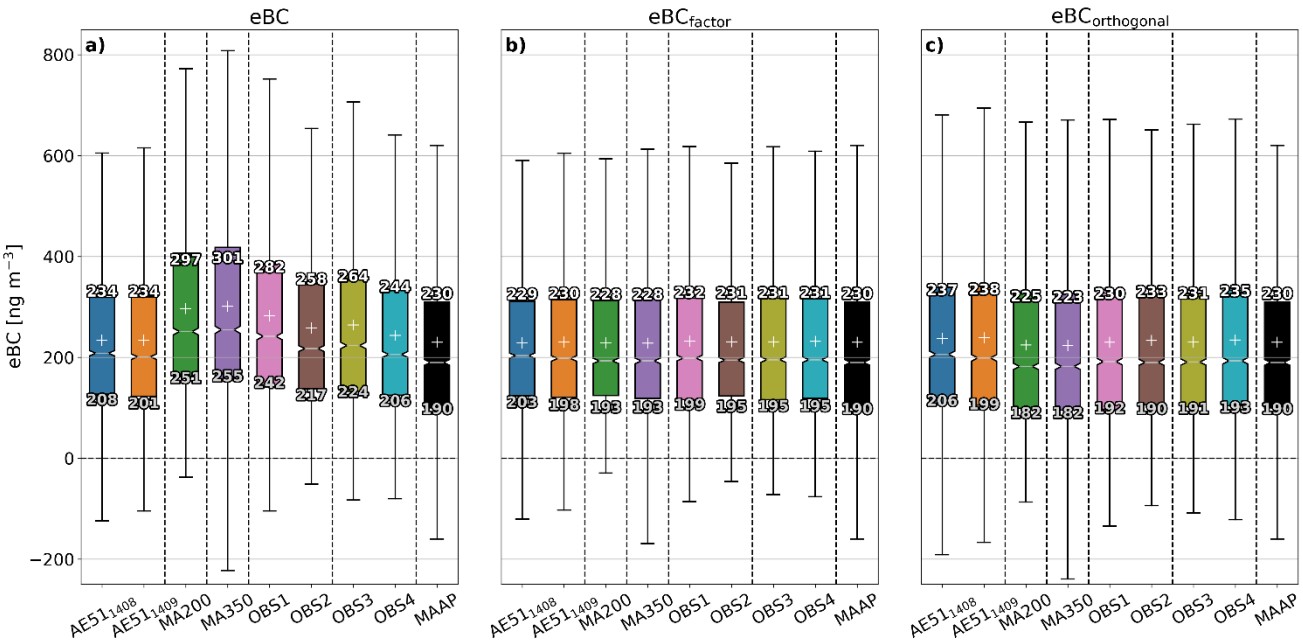

**Figure 5**. *Calibration methods: a) data without calibration; b) data calibrated by the F-factor calculated by comparing 1$^{st}$ intercomparison data means; and c) data calibrated with the orthogonal fit equations. In the boxplots the middle line shows the median, "+" shows mean, top of box 75$^{th}$ percentile, bottom of the box 25$^{th}$ percentile and top and bottom whisker the last points within 1.5 times the interquartile range. The values are the mean (top) and median (bottom).*

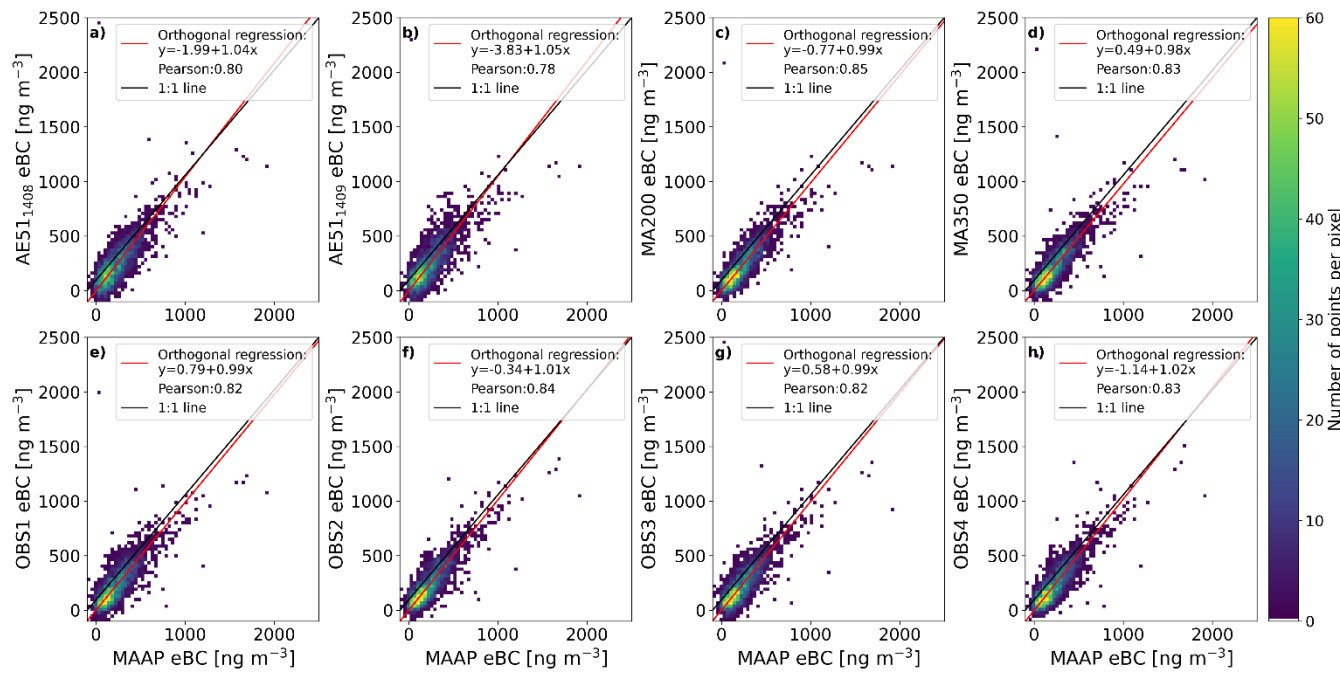

**Figure 6**. *Correlation and orthogonal regression line fits after calibrating with the orthogonal regression equations presented in Fig. 3 and Table 4. Data are from 1st intercomparisons 26.5.–6.6.2022 as 5-min averages.*

**Table 5.** *Table of the effects of the calibration with the orthogonal regression.*

| | 1st Intercomparison 26.5–6.6.2022 | | | 1st Intercomparison 1 26.5–6.6.2022 after calibration | | |
|---|---|---|---|---|---|---|
| Sensor | Intercept | Slope | $r$ | Intercept | Slope | $r$ |
| AE51$_{1408}$ | 42.1±2.81 | 0.84±0.01 | 0.80 | -1.99±3.47 | 1.04±0.01 | 0.80 |
| AE51$_{1409}$ | 41.9±3.01 | 0.84±0.01 | 0.78 | -3.83±3.73 | 1.05±0.01 | 0.78 |
| MA200 spot1 | 51.5±3.06 | 1.08±0.01 | 0.85 | -0.77±2.81 | 0.99±0.01 | 0.85 |
| MA350 spot1 | 42.6±3.43 | 1.13±0.01 | 0.83 | 0.49±2.97 | 0.98±0.01 | 0.83 |
| OBS1 | 36.0±3.38 | 1.06±0.01 | 0.82 | 0.79±3.15 | 0.99±0.01 | 0.82 |
| OBS2 | 39.5±2.87 | 0.95±0.01 | 0.84 | -0.34±3.06 | 1.01±0.01 | 0.84 |
| OBS3 | 27.8±3.29 | 1.02±0.01 | 0.82 | 0.58±3.20 | 0.99±0.01 | 0.82 |
| OBS4 | 33.5±2.79 | 0.91±0.01 | 0.83 | -1.14±3.12 | 1.02±0.01 | 0.83 |

## 3.2 Temporal and spatial variability during deployment

### 3.2.1 General features and spatial variability of *eBC* in Kumpula

Figure 7 presents the time series of the whole deployment period (4.7.–16.9.2022) for all the sensors. The two-week period ( 4.7.–19.7.2022), when all the sensors were operational (the $F_{roof}$ sensor stopped working in 19.7.), is marked in Fig. 7b, and a comparison during this period can be seen in Fig. 8a. All locations had a statistically significant difference (Fig. A2.1), although the differences were not necessarily remarkable.

The $F_{roof}$ and $F_{ground}$ locations had the lowest concentrations and the highest concentrations of *eBC* were measured at $P_{roof}$ and $P_{ground}$, respectively. At $P_{roof}$ and $P_{ground}$, multiple short-term high-concentration peaks were observed possibly caused by the proximity of the bus stop. The bus stop has approximately 160 buses stopping on it per day with the peak during the day having 9 to 12 buses per hour. $BG_{ground}$ showed similar median concentration to the $P_{roof}$ and $P_{ground}$ locations, but without the local source peaks at the $P_{roof}$ and $P_{ground}$. At $SMEARIII_{ground}$, we observed slightly higher concentrations than at the closest site $F_{ground}$ but lower than the $P_{roof}$ and $P_{ground}$ locations. The local source peaks for MAAP at $SMEARIII_{ground}$ were in between the magnitudes of the respective $F_{roof}$ and $F_{ground}$ to $P_{roof}$ and $P_{ground}$ values. Minimal vertical difference in the eBC concentration was observed between the $P_{roof}$ and $P_{ground}$ locations. Similarly, at $F_{roof}$ and $F_{ground}$, respectively, minimal vertical difference was observed.

When considering the whole deployment period (Fig. 8b), two distinct areas could be identified. The locations closer to the Kustaa Vaasa road of $P_{roof}$ and $P_{ground}$ and the further away backgrounds of $F_{ground}$, $SMEARIII_{ground}$ and $BG_{ground}$. The difference between the areas is perhaps traffic proximity due to the Kustaa Vaasa road and bus traffic past the $P_{roof}$ and $P_{ground}$ locations. This causes $P_{roof}$ and $P_{ground}$ to measure approximately 50 ng m$^{-3}$ higher concentrations. The difference is relatively negligible as the instrument precisions are in the same magnitudes and the ambient conditions are challenging for the sensors.

The overall concentrations were lower than in previous studies conducted in the Helsinki region. Luoma et al. (2021b) reported annual means of 510 to 530 ng m$^{-3}$ at urban background cites. Helin et al. (2018) reported average±standard deviation concentrations of $1940 \pm 1530$ ng m$^{-3}$ at a street canyon and $450 \pm 420$ ng m$^{-3}$ at a detached residential area in the summertime of 2016. The concentrations were similar with corresponding values of urban background cites during summers of 2017-2019 within Northern Europe (~240 – 340 ng m$^{-3}$) and lower than in Western and Central Europe (~330 – 1480 ng m$^{-3}$) (Table S2, Savadkoohi et al., 2023).

In comparison toCaubel et al. (2019), who operated 100 *eBC* sensors for 100 days in a borough sized area, we observed only small variations within the much smaller campus area. Caubel et al. (2019)reported considerably larger differences between the sensor locations: for example, 200-400 ng m$^{-3}$ in upwind locations that were less affected by the anthropogenic activities,

and 500-1200 ng m$^{-3}$ in a busy port environment. Residental concentrations were reported to be on average slightly higher 400-500 ng m$^{-3}$ in comparison to the 250-400 ng m$^{-3}$ measured in this study.

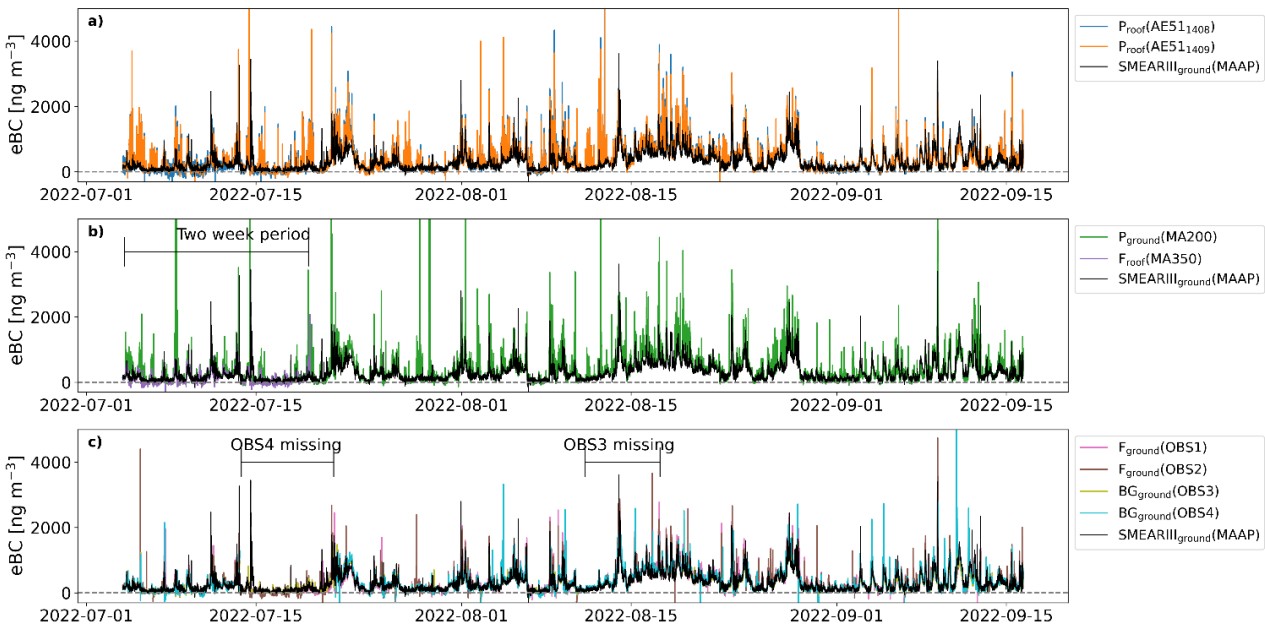

**Figure 7**. *Time series of eBC for the deployment period for a) AE51 sensors, b) MA-series sensors, and c) Observair (OBS) sensors. Each panel has also the SMEARIII$_{ground}$ measurement by the multi-angle absorption photometer (MAAP) reported. Data are in 5-min averages.*
*The two week period, when all the instruments were operational is marked in panel b) and the periods when OBS3&4 sensors were malfunctioning are marked in panel c).*

Sources of the BC were studied with a wind rose analysis shown in Fig. 9. The wind roses for different locations mostly tell a similar story: highest *eBC* concentrations were measured with low wind speeds especially blowing from the east, when the *eBC* was transported to the campus area from the busy road (Kustaa Vaasan tie). The low wind speeds were also tied to the
evening times with the accumulation of pollutants due to the more stable atmosphere.

The effect of the nearby construction site was not clearly visible in the data. Only at the P$_{ground}$ and P$_{roof}$ locations there were some increased concentrations from the direction of the construction site (south-west). For SMEARIII$_{ground}$ or F$_{ground}$, the direction of the construction site (north-east) did not stand out. At SMEARIII$_{ground}$, increased concentrations on higher wind speeds from west observed, which is probably caused by a single pollution event and was captured due to the higher inlet
location. For P$_{roof}$ this direction is also shielded by the building where this location resides. Similar results have also been reported by Alas et al. (2019) who did not observe increased *eBC* concentrations close to construction sites.

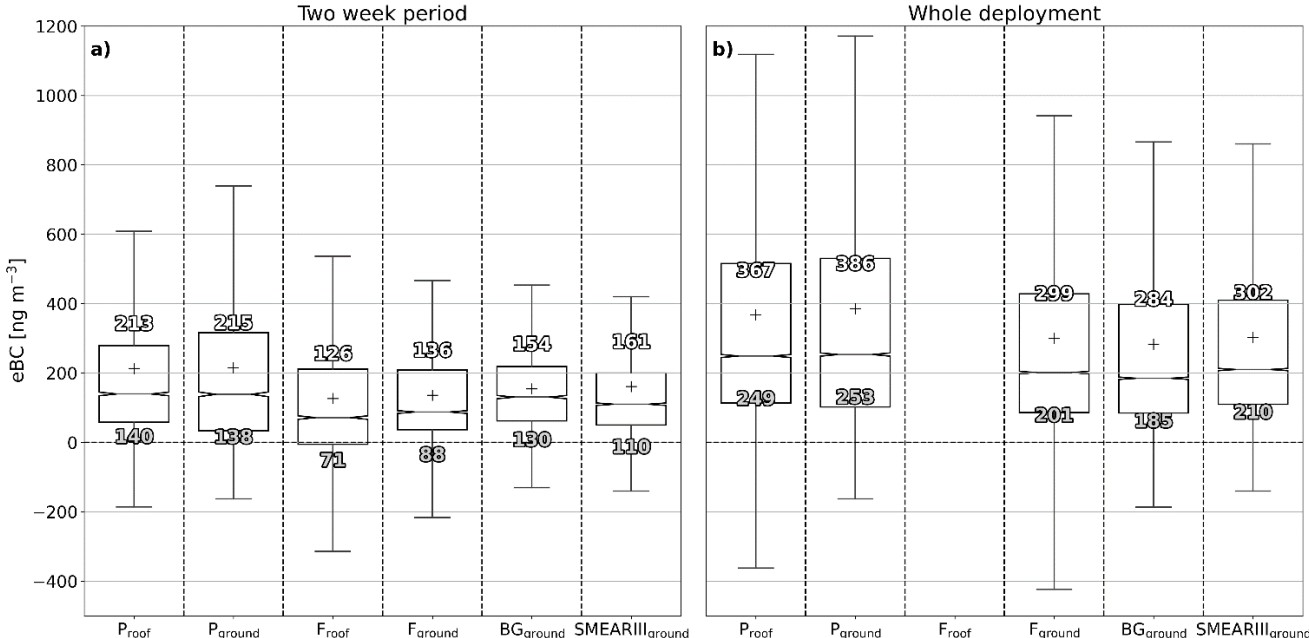

**Figure 8**. *Boxplots of from the deployment period: a) data are only from the first 15 days (4.7.–19.7.) of the deployment when all the instruments were operational; and b) data are from the whole deployment phase (74 d). The explanation for the boxes in the same as in Fig. 5.*

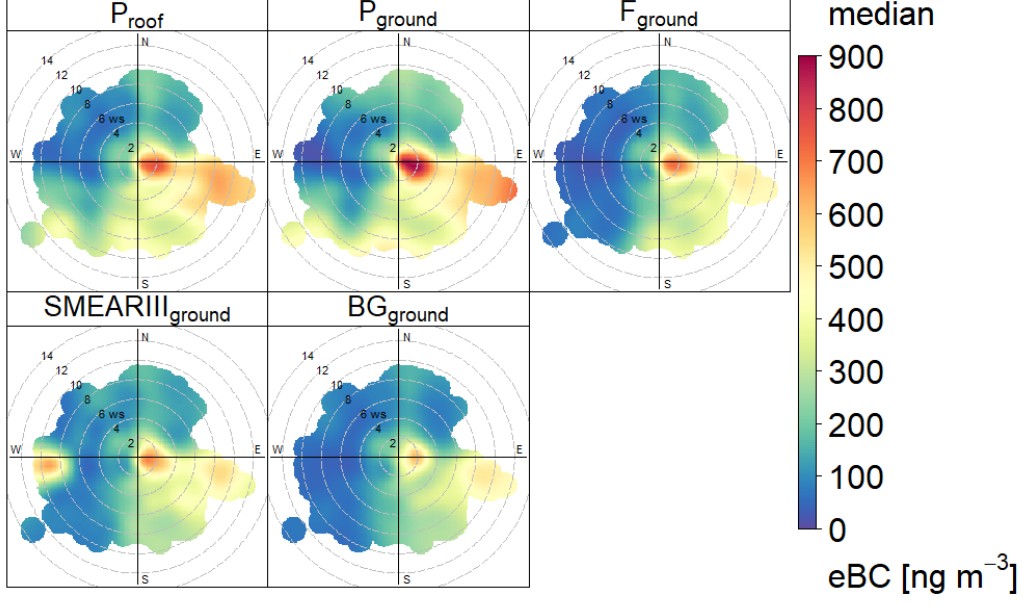

**Figure 9.** *Wind roses of the deployment phase showing median eBC concentration measured with different sensors as function of wind speed (WS, in units of [m s⁻¹]) and direction.*

### 3.2.2 *eBC* concentration during days of the week

A daily breakdown can be seen in Fig. 10. There is somewhat surprising variation on day-to-day basis, as no notable differences were expected between weekdays. At all the locations, Mon and Tue had statistically significantly (Fig. A2.2) higher concentrations than Wed and Thu. The most stable locations were $F_{ground}$ and $SMEARIII_{ground}$, where there was no statistically significant difference between Mon, Tue, Fri, Sat and Sun. Therefore, weekend and weekdays did not seem to have a clear difference in the medians to each other, which is differing compared to other studies that observed lower *eBC*

concentrations during weekends at traffic and at urban background sites in Helsinki (Helin et al., 2018; Luoma et al., 2021b). Also, Caubel et al. (2019) reported lower concentrations during weekends especially on traffic influenced cites. It is to be noted that the variance in concentrations was higher during Fri–Sun than during Mon–Thu and the highest peaks were measured during the weekend. The unexpected similarity between the weekdays and weekends might be due to a rather short period (74 d) for such an analysis and the time of the deployment period, which is a vacation season in Finland, when the

anthropogenic activities are expected to depend less on the days of the week.

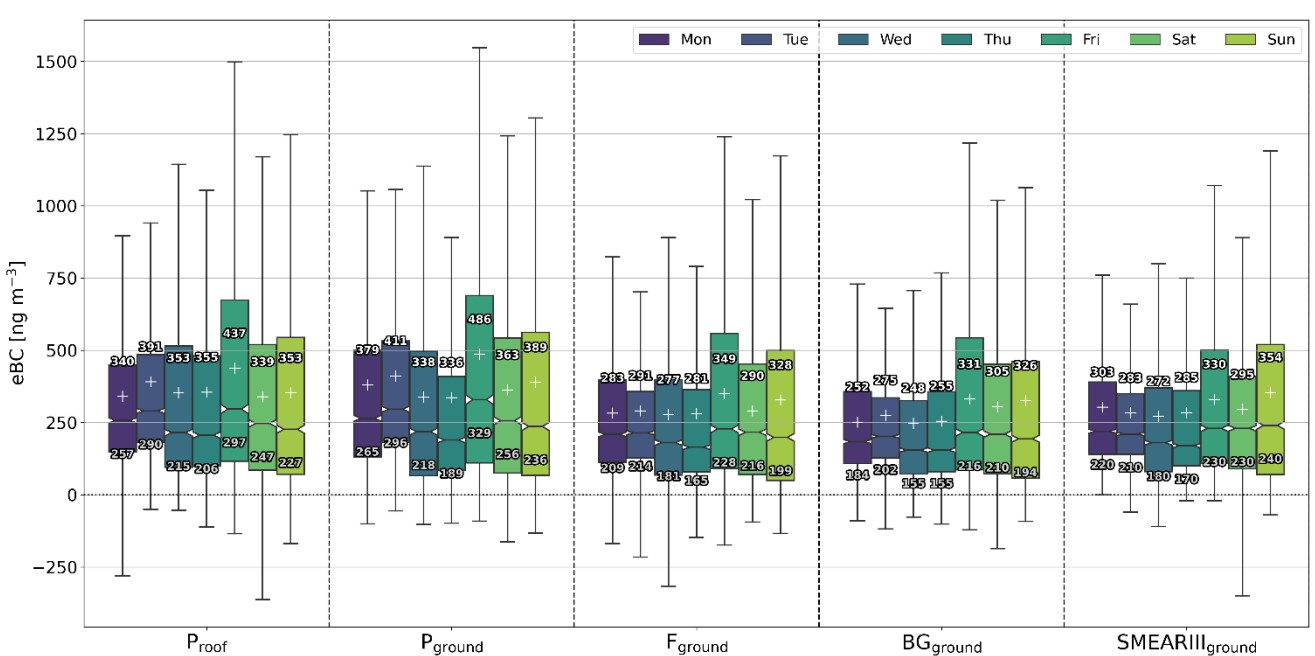

**Figure 10.** *Daily eBC concentrations for different sensors. In the boxplot every sensor has 7 boxes going left to right as Mon–Sun (indicated with different colors). The explanation for the boxes in the same as in Fig. 5.*

$P_{roof}$ (AE51) filters were most commonly changed Mon and Fri and $F_{ground}$ and $BG_{ground}$ (Observairs) Mon–Wed with only

exception Friday 19.8. With the single filter instruments the significant loading effects should be considered as a pattern of data collection behavior could implicate false patterns of *eBC* in the daily variability. However, a rather similar day-to-day pattern is observed at all the different sites, even at $SMEARIII_{ground}$ and $P_{ground}$, where the filter was changed automatically at

random periods. Therefore, we can conclude that the weekday variation seen in the *eBC* concentrations was not remarkably influenced by the filter changing cycles.

### 3.2.3 Diurnal variation in BC concentration

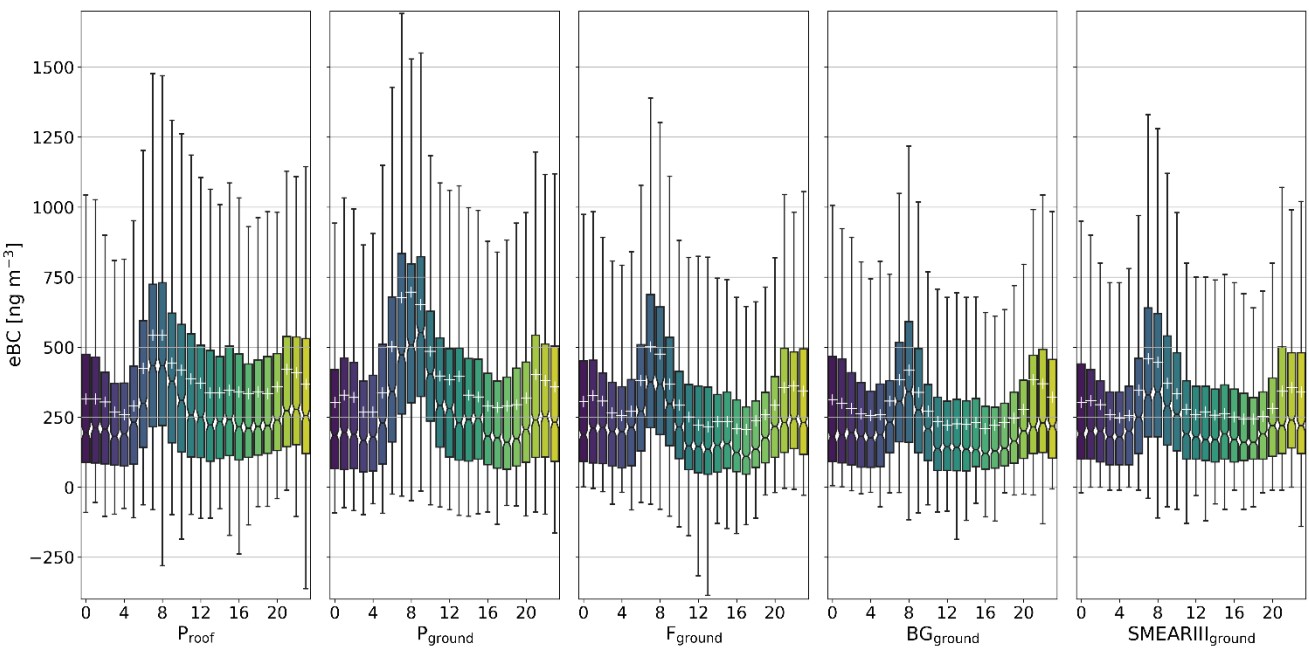

**Figure 11.** *Hourly variation of eBC concentrations. In the boxplot every sensor has 24 boxes going from 00–23 where the box describes the hour of the day. The explanation for the boxes in the same as in Fig. 5.*

The diurnal variation of *eBC* can be seen in Fig. 11, which shows a similar diurnal pattern at all the locations. The variation is affected by the local and regional anthropogenic activities and meteorological conditions. The *eBC* concentrations sharply rose during the morning due to increase in traffic. The highest concentrations were reached between 9–10 after which the concentrations decreased due to smaller traffic rates, increased dilution in the convective boundary layer due to higher mixing height and increased wind speeds (e.g., Fig. S2 in Luoma et al., 2021b). Another rise in concentration was observed late in the evening around 21–23. This increase was much less compared to the morning peak. The increased levels during the evenings are probably caused by accumulation of pollutants in a more stable atmosphere when the mixing height is lower and the wind speeds are also generally lower. Based on observations made in Helsinki, (Järvi et al., 2009)reported that out of the meteorological parameters, the wind speed and mixing height had the greatest effect on *eBC* concentrations. Also, local wood combustion emissions, for example, evening activities at the close by community garden, can increase the *eBC* levels. Similar diurnal patterns with a peak in the morning and evening have been observed by previous studies during the warm period at traffic and urban background sites (Sahu et al., 2011; Backman et al., 2012; Caubel et al., 2019; Luoma et al., 2021b).

### 3.2.4 Artifacts caused by sensor overheating

During the measurements overheating of sensors was observed in all locations utilizing the weatherproof boxes ($P_{ground}$, $F_{roof}$, $F_{ground}$, $BG_{ground}$). This was due to the increase of ambient temperature after sunrise and in some locations direct sunlight
heating the black weatherproof boxes.

With the MA-series sensors (MA200, MA350) the change of the $T$ and $RH$ caused clearly erroneous data as seen in Figs. 12 and 13. According to the instrument manual, the operating temperature for the instrument is 0-40 °C. In our deployment, the T increases even to 52 °C, which is above the operating temperature and could explain the behaviour. However, the anomalous activity is observed even below the 40 °C. Previous studies have shown that sharp changes in $T$ and $RH$ can
cause positive or negative spikes in the measurement of filter-based optical methods (Caubel et al., 2018; Düsing et al., 2019). The reason for this artifact is considered to be mostly influenced by the detector, LED properties and other electronics affected by the $T$ change and sorption and desorption of the filter fibers due to changing $RH$. The largest error in the measurement is when the $T$ change was the fastest, around 9–11 in the morning. The dualspot correction was observed to amplify the measurement error of the individual spots.

For the Observair sensors ($F_{ground}$ and $BG_{ground)}$ the influence of overheating was negligible when compared to $SMEARIII_{ground,}$ due to the automatic environmental compensation algorithm used in the sensors described in Caubel et al. (2018).

The large overall change in $T$ most likely caused a strain on the pumps reducing the lifetime of the sensors. This may have contributed to the failure of the MA350 sensor pump during deployment. With AE51, at the $P_{roof}$, no problems related to $T$
and $RH$ were observed due to the deployment location being inside in a controlled laboratory space, but similar behavior could be expected if these sensors are deployed in ambient conditions.

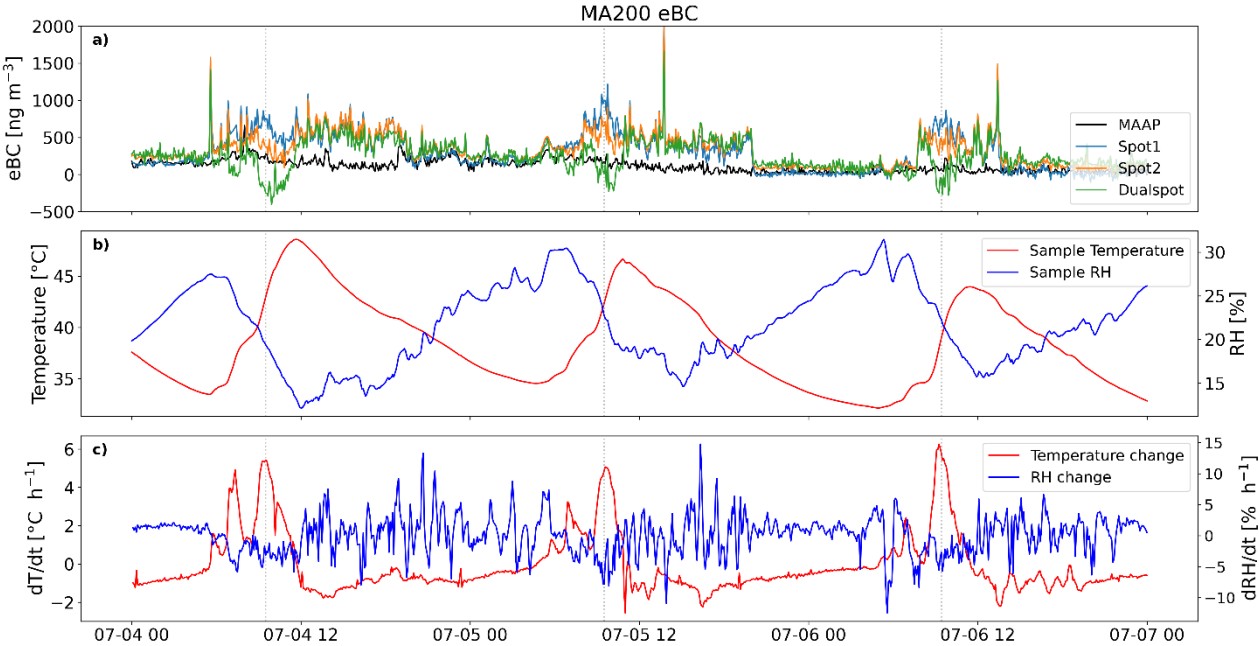

**Figure 12.** *Example of the MA200 T/RH artifact as time series of a) eBC, b) temperature (T) and relative humidity (RH), and c) the T and RH change rate (dT dt⁻¹ and dRH dt⁻¹, respectively). All data are 5-min averages.*

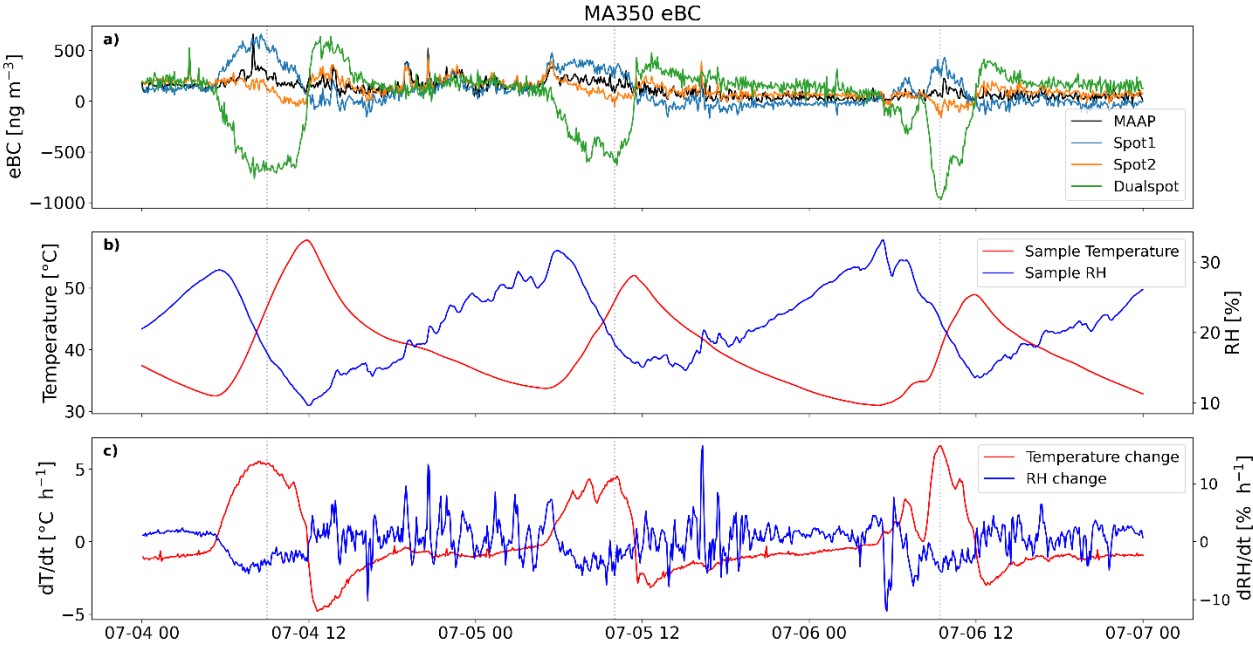


**Figure 13.** *Example of the MA350 T/RH artifact as time series of a) eBC, b) temperature (T) and relative humidity (RH), and c) the T and RH change rate (dT dt⁻¹ and dRH dt⁻¹, respectively). All data are 5-min averages.*

## 4 Conclusions

In this study, four different types of *eBC* sensors were used as a sensor network firstly to study variation of *eBC* in urban environment and secondly to study applicability of *eBC* sensors to monitor ambient BC concentrations in real conditions. The results were compared to reference level instrument results to validate the results.

During the intercomparison periods, the correlations between different *eBC* sensors and the reference instrument were good ($R \approx 0.8$, 5-min averages) but the slopes of the regression lines varied from 0.8 to 1.1 indicating a need for sensor-specific calibration. The *eBC* sensors observed the temporal variation well and the *eBC* levels varied according to anthropogenic activities in the local and regional area (e.g., in the nearby busy road), and meteorological conditions. For the spatial variation we observed only small variation. Surprisingly, the local construction site, which was assumed to cause an increase in *eBC* data, did not stand out in the results. Due to the lack of local emission sources in the studied area, the variation in *eBC* in an urban background location was observed to be minimal. Based on our results, the reference scale SMEAR III station, which is classified as an urban background site, represents the pollution levels in the campus area well. Taking the sensor network closer to local anthropogenic sources (e.g., right next to a busy road), the gradients of *eBC* concentration are expected to be more remarkable.

Due to their small size enabling easy installation to existing structures (like sheds or roofs) and affordability the sensors were observed to be well suited to building a sensor network in an urban area. However, still in field conditions, several issues were observed. The performance of the dualspot correction should be evaluated before field campaigns for small scale sensors that have the capability for this correction. Due to the small size and much lower flow rates the sensors show significant instability in the determination of the correction parameter *k* with the available methods. In this study during the intercomparison measurements with the dualspot corrections were unstable in temperature controlled environment. During deployment measurements changes in temperature caused additional errors in the measurements of the individual spots which were amplified by the dualspot correction. This effect is especially important with sensors like the MA200 and MA350, which by default give the measurement result as the dualspot corrected data.

Temperature changes significantly affected the measurements and provided a challenge in the deployment of the sensor network. Development of robust enclosures or deployment in locations that have stable or controlled temperature is needed. Alternatively, the environmental compensation used by the Observair sensors was seen to reduce the effect of temperature changes. Therefore, a suggestion is made that the environmental compensation utilized by the Observair and outlined in Caubel et al. (2018) could be applied as a measurement method to the data via post-processing or implemented to other sensors by manufacturers as a solution to the temperature artifacts.

It is not possible to say which sensor performed the best as the sensor design differs significantly. The user needs to take into account the requirements of the measurement environment and the features of the individual sensor types (number of

wavelengths, filter capacity and other maintenance needs, and price). In our conditions sensors performed near equally if single spot data is used for the MA-sensors. Observair performed slightly more stable in changing conditions. Comparing the instrument performance, it is to be noted that AE51 was run in temperature-controlled environment while Observair and MA-series sensors were exposed to varying temperature in the deployment boxes. With AE51 and Observair the filter change needs to be done every few days whereas MA-sensors can measure independently for months. MA-sensors also offer the wavelengths for estimating *BrC* concentration, which were not utilized in this study.

*Code availability*

Gitlab: https://version.helsinki.fi/elomaata/UAQ2-0

*Data availability*

Gitlab: https://version.helsinki.fi/elomaata/UAQ2-0

*Author contributions*

TE did the data analysis and main part of the writing with the help by KL. TE and SH were the main responsible people to build the sensor network and keep it running. All the authors contributed to the planning of the study, interpreting the data, and commenting the manuscript. HT and TP supervised the work and organized the funding.

*Competing interests*

At least one of the (co-)authors is a member of the editorial board of Aerosol Research.

*Acknowledgements*

Financial support from Urban Air Quality 2.0 project funded by Technology Industries of Finland Centennial Foundation, H2020 project RI-URBANS (Grant agreement No 101036245), the Academy of Finland via the project Black and Brown Carbon in the Atmosphere and the Cryosphere (BBrCAC) (decision nr. 341271) and Academy of Finland Flagship funding

(grant no. 337552, 337549) are gratefully acknowledged

## A1. Flow calibrations during the campaign

The sensor flow rates were calibrated before the measurements (on 25.5.) with a Alicat Scientific M-series mass flow meter. The calibration was done manually according to the operating manuals for the AE51s and Observairs and the automatic flow calibration program was used for the MA-series sensors. For OBS1 the flow calibration was $\pm 2$ ml min$^{-1}$ and for the other AE51 and OBS sensors $\pm 1$ ml min$^{-1}$. The MA-sensors passed the automatic calibration program. OBS1 and 2 flows were checked after the 1$^{st}$ intercomparison. MA350 was flow calibrated on 9.8. All OBS flows were calibrated on 19.8 and results were within $\pm 1$ ml min$^{-1}$. On 30.8 all sensors were flow calibrated. AE51s were within $\pm 1$ ml min$^{-1}$ and OBSs were within $\pm 2$ ml min-1. MA200 flow calibration failed, and the flow given by the instrument in relation to the flow meter was $+4$ ml min$^{-1}$. Also, during the calibration AE51$_{1408}$ could not reach the maximum flow of the pump of 250 ml min$^{-1}$ therefore showing fatigue and deterioration of the pump. The results of flow calibrations are collected to Table A1.

**Table A1**. *Flow calibrations of the BC sensors during the measurements*

| Sensor | 25.5 | 6.6 | 9.8 | 19.8 | 30.8 |
|---|---|---|---|---|---|
| AE51$_{1408}$ | $\pm 1$ ml min$^{-1}$ | | | | $\pm 1$ ml min$^{-1}$, could not reach max 250 ml min$^{-1}$ |
| AE51$_{1409}$ | $\pm 1$ ml min$^{-1}$ | | | | $\pm 1$ ml min$^{-1}$ |
| MA200 | passed | | | | Failed, $+4$ ml min$^{-1}$ |
| MA350 | passed | | passed, was blocked before from 22.7., irregularities from 19.7. | blocked after data loading, tube was loose fixed 30.8. | Tube fixed, no flow adjustments, data unusable |
| OBS1 | $\pm 2$ ml min$^{-1}$ | checked | | $\pm 1$ ml min$^{-1}$ | $\pm 2$ ml min$^{-1}$ |
| OBS2 | $\pm 1$ ml min$^{-1}$ | checked | | $\pm 1$ ml min$^{-1}$ | $\pm 2$ ml min$^{-1}$ |
| OBS3 | $\pm 1$ ml min$^{-1}$ | | | $\pm 1$ ml min$^{-1}$ | $\pm 2$ ml min$^{-1}$ |
| OBS4 | $\pm 1$ ml min$^{-1}$ | | | $\pm 1$ ml min$^{-1}$ | $\pm 2$ ml min$^{-1}$ |

**A2. Statistical significance during deployment**

During the deployment statistically significant differences were observed between all locations as seen in Fig. A2.1. The highest p-value is observed between SMEARIII and $F_{ground}$, which are the closest proximity sites at similar height. This value is still significantly lower than 0.05. The statistical significance between weekdays for every location is presented in Fig. A2.2.

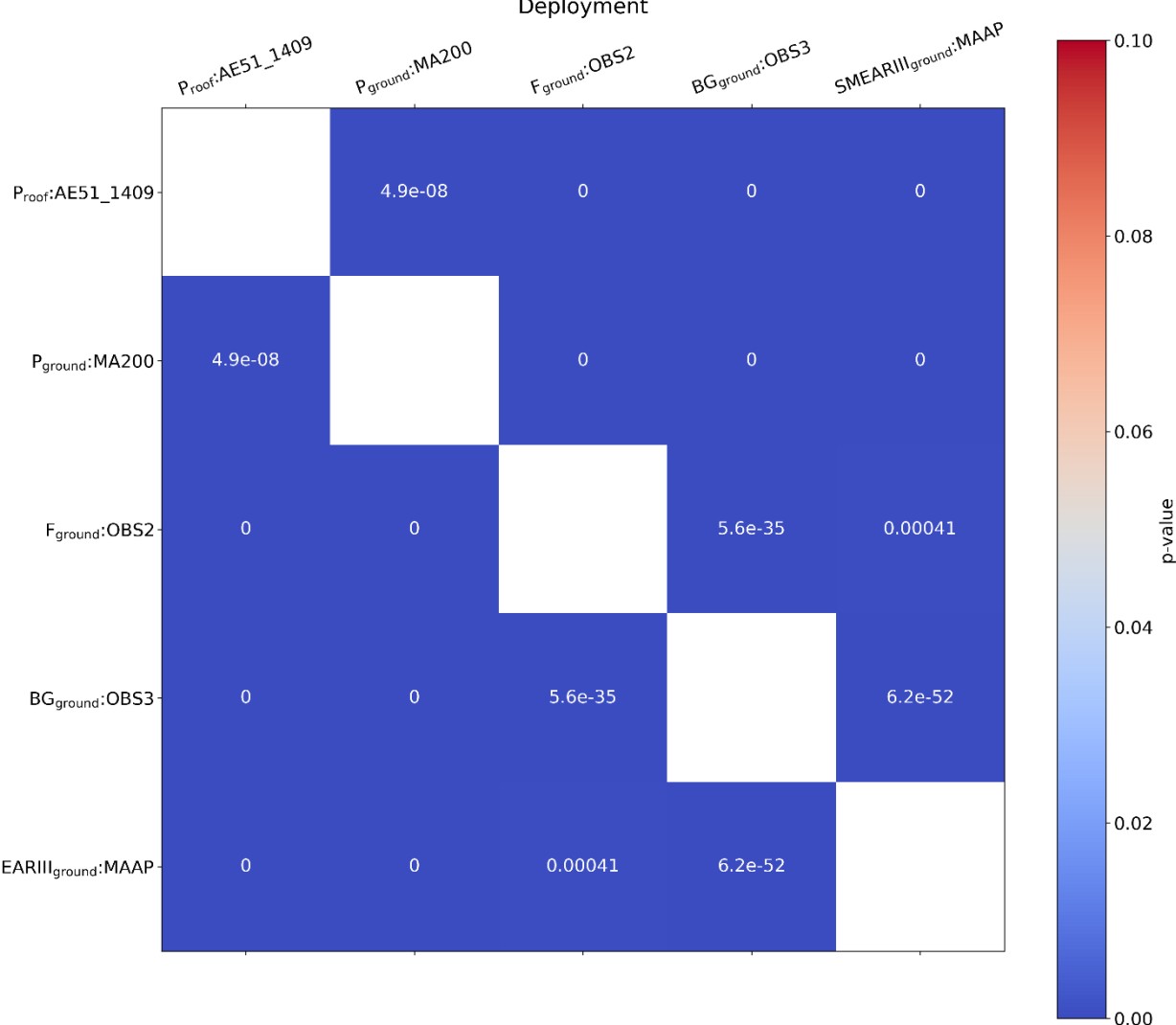

**Figure A2.1**. *Statistical significance between the locations during the deployment phase. The p-values are calculated with the Willcoxon signed-rank test.*

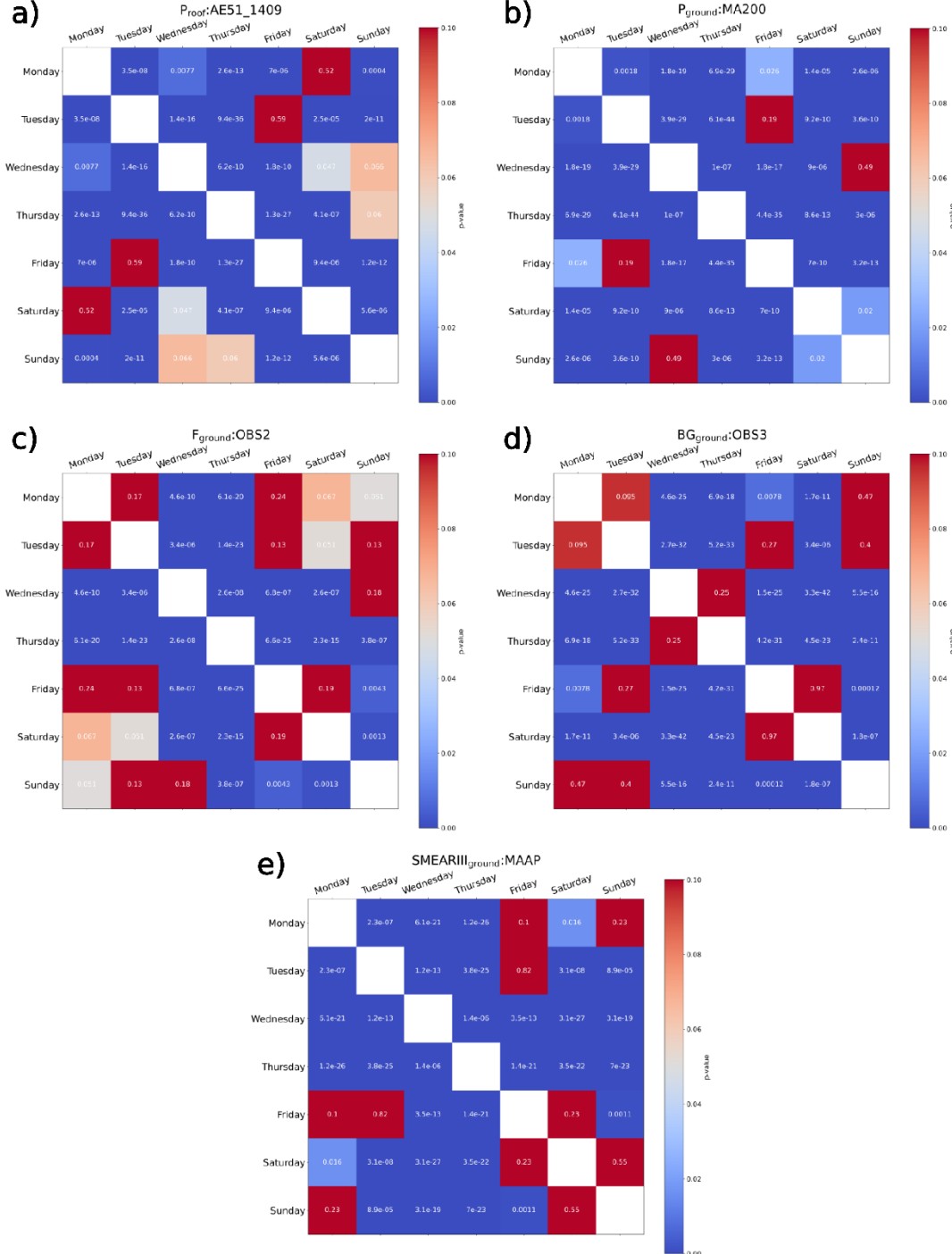

**Figure A2.2**. *Statistical significance between the weekdays during the deployment phase for every measurement location. The p-values are calculated with the Willcoxon signed-rank test.*

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
