# Peer review of "The applicability and challenges of black carbon sensors in monitoring networks"

_Aerosol Research, 2024_

## Author Response (AR1)

**Referee #1**

We thank the Referee #1 for the valuable comments on the manuscript. Please find the point-to-point answers to the comments below.

**• Careful checking of the English language and the text as a whole is necessary.**

The manuscript has been proofread in order to check for any mistakes in grammar and spelling.

**• There are also a few misstatements which I would advise correcting.**

Removed statement that "Black Carbon is the second highest warming agent in the atmosphere" as the newest IPCC report does not support this statement.

**• The author should change the title. It does not only examine the spatial variability of ambient black carbon.**

The title has been changed to:

The applicability and challenges of black carbon sensors in dense monitoring networks

**• In introduction: Please correct PM2.5 to PM2.5.**

Corrected.

**• Page 11, Paragraph 210: A sharp ATN change |ATN| >30 was manually identified. What could be the reason for this sharp change? Please explain it.**

The original wording was indeed confusing and referred to the sharp change of ATN related to the filter change. As an example, if a filter has an ATN of 80 and it is changed, the new filter will have a ATN of 0. Therefore, the change in ATN is larger than 30 i.e  $|\Delta ATN| > 30$

The wording has been changed to: The filter changes were manually identified, and two hours of data were removed starting from the nearest hour before the filter change.

**• Page 11, 3.1 section: The authors used different flow rates for the same type of sensors. This is sometime hard to follow in the article. Please create a table summarizing this. How much during each campaign, etc.**

Table 3 has been added at line 210 to summarize the flow rates used at specific date ranges with the specific sensors. A reference to the table has been added to the start of paragraph 190. Section 3.1 has been edited to prefer referring to Table 3 when necessary.

• Page 12, Paragraph 255: The authors wrote that the 1st intercomparison has on average lower concentrations compared to the 2nd intercomparison. Is it not because of the different meterological condition? Please explain it.

Yes, the difference is most likely because different meteorological conditions between these periods. Also, the 1st intercomparison is at the start of the summer vacation season in Finland when traffic density can be expected to be less in the urban environment. In comparison the 2nd intercomparison is more everyday life when traffic especially on the Kustaa Vaasa road can be expected to be quite heavy.

Additions have been made to the statement as follows:

The 1st intercomparison has on average lower concentrations compared to the 2nd intercomparison. This is due to the difference in meteorological conditions and in traffic density during these periods.

**• Page 13, Figure 2: Please check the labels (date) at the xaxis. Please correct it.**

The figure features a split x-axis marked with the red vertical lines and the dashes on the axis itself. I.e it consists of  $1^{st}$  Intercomparison on the left, missing section of approximately 3.5 months in the middle and then the  $2^{nd}$  Intercomparison. The labels are therefore accurate in their representation. The figure caption has been updated to:

**Figure 2.** Timeseries of both intercomparison periods a) 26.5.-6.6.2022 and b) 16.9.-3.10.2022. In the figure there is a split x-axis, where the period in between panels (a) and (b) marked with the vertical red lines is approximately 3.5 months. This period was the deployment phase between the intercomparisons. Data points are 5-minute averages.

• Page22, Paragraph 400: The authors wrote the following: With the MA-series sensors (MA200, MA350) the change of the temperature and RH caused clearly erroneous data as seen in Fig. 12. However, we cannot see the results of MA200 sensor in the Figure 12. Please include its results in the figure.

Figure 12 has been renamed to Figure 13 and Figure 12 has been added to show the results of the MA200. A separate figure has been used to ensure the clarity of the figures.

**• The conclusion contains some statements that need to be clarified. For istance: what is DST? please explain it.**

Added the abbreviation DST to line 136 at page 6 (Distributed Sensing Technologies). Unfortunately, Distributed Sensing Technologies ceased operations in the end of 2023. Therefore, the conclusion paragraph has been revised to remove any mentions of the company and instead refer to the Observair sensor. For the environmental compensation, reference to the original publication Caubel et al. (2018) is added and a clarifying statement as the original publication uses the name Aerosol Black Carbon Detector (ABCD) for the Observair sensor. The paragraph reads now:

Temperature changes significantly affected the measurements and provided a challenge in the deployment of the sensor network. Development of robust enclosures or deployment in locations

that have stable or controlled temperature is needed. Alternatively, the environmental compensation used by the Observair sensors was seen to reduce the effect of temperature changes. Unfortunately, the Observair sensors are not being produced as of the end of 2023. Therefore, a suggestion is made that the environmental compensation utilized by the Observair and outlined in Caubel et al. (2018) could be applied as a measurement method to the data via post-processing or implemented to other sensors by manufacturers as a solution to the temperature artifacts. Please note that in the publication Caubel et al. (2018) the name Aerosol Black Carbon Detector (ABCD) is used, which is the academic prototype of the Observair sensor.

**Referee #2**

We thank the Referee #2 for the valuable comments on the manuscript. Please find the point-to-point answers to the comments below.

**1.** A language check is necessary for the entire manuscript. Some segments of the text do not follow common practices of reporting data or manuscript writing, such as consistently reporting units of mass concentration or time.**

The manuscript has been checked in order to catch any deviancies from the guidelines.

**2.** Section titles require renaming (based on the updated context), particularly in Section 2 (Methods) and Section 3 (Results).**

Following section titles have been updated. Section 2.2 "Dualspot correction algorithms". 2.4 "Deployment area". 3.1 "Intercomparison periods". 3.1.1 "Applicability of the dualspot corrections". 3.2.2 "Weekly features in BC concentration"

**3.** I highly recommend adding more text discussing the results and making section **3** from "Results" to "Results and Discussions," which is a typical nature of manuscript framing in similar types of scientific journals.**

The title has been adjusted to the suggestion.

**4. The title of the manuscript can be updated as the manuscript content does not presently match with the title.**

The title has been changed to:

The applicability and challenges of black carbon sensors in dense monitoring networks.

**5. Use PM2.5 everywhere**

Corrected.

**6. Line 71: Why is MAAP chosen as a reference device to compare with Aethalometers, which work slightly differently? AE33 could be a great reference device in this work. If**

**the authors have AE33's data, adding a comparison might help understand the unitspecific offsets in measuring BC. If they don't, I highly recommend providing sufficient justification (here in Line 71) or in the methods section why MAAP is used or better suitable.**

Unfortunately, no AE33 data was available during this campaign. MAAP was utilized as the reference instrument as it was the highest-grade instrument available during this period.

**7. Line 85: Some theoretical mistakes were identified. Please check the literature and correct it. Aethelometers measure light intensity and calculate light attenuation (ATN). From ATN measurements, ATN coefficients (bATN) are derived, followed by the absorption coefficient (babs). Drinovec 2015 explained this well.**

Error fixed. "The measured variable by the instrument is the attenuation coefficient  $b_{atn}(\lambda)$  [m-1] calculated from the measured attenuation and the operational parameters of the instrument as described in Eq. 2."

**8. Line 105: There are inconsistencies around the assumptions made. If any scattering correction is made, why are they assumed to be unity? More explanations are required. Also refer to Line 152.**

Scattering correction is the  $s(\lambda)\sigma_{sp}(\lambda)$  part of the general correction scheme (Eq. 3). This refers to the scattering from the aerosols and is ignored in this study which was meant by saying "assumed unity". The multiple scattering correction (Cref) refers to the enhanced attenuation due to the filter fibers. For this the manufacturer recommendations are used. This statement is rephrased to:

"The aerosol scattering correction requires measurement of the scattering coefficient, which in many cases is not possible due to the lack of instrumentation. Due to this the aerosol scattering correction is often voided as in this study"

Text has been adjusted that in reference to these to corrections, aerosol scattering correction is used for the  $s(\lambda)\sigma_{sp}(\lambda)$  part and multiple scattering correction for the Cref.

**9. Line 118: Leakage factor changes over time. Please refer to Drinovec 2015. This assumption might change the corrections, and so might the final corrected BC concentration. Please reconsider adapting to such changes.**

The line referring to the leakage factor has been removed from the text and equations. This was an error in the manuscript. The leakage factor was tested during data processing of the manuscript but later removed.

**10. It is not clear if the MA series device's inbuilt Dual-spot corrections were used for comparison or not. Typically, MA devices have their own correction mechanism, which is not the same as Drinovec 2015. Please check and confirm with a table of corrected data or add it to Table 2.**

The MA series inbuilt corrections are used. Meaning the dualspot corrected data that the instrument gives is used as is.

Confusion to this is related that the authors were not aware of the exact correction algorithm used by the MA-series instruments. It was assumed that Dualspot refers to the correction outlined in Drinovec et al. (2015). This is now considered and refered in the modified version of the manuscript.

**11.** Some comments on figures have been mentioned in the attached file, which mainly focuses on the visibility of the graphics and texts.**

These comments have been taken into account and the plots updated.

12. Section 3.1.2: A separate segment on the sensor calibration in methods sections is recommended. Also, mention how the calibrations are assessed (metrics used, such as slope, MAE). Reporting the calibration results are helpful for future studies. While presenting the data in "Result" section, please compare it with previous literature. A table might be helpful for reporting metrics from uncalibrated data, calibrated data, and literature data - with the type of calibration procedure adopted. All these elements will expand this section, which I believe is going to improve the quality of the manuscript.

Calibration methodology is added to section 2.5. The results of the calibration are showed in section 3.1.2

13. Generally, the results and discussion section is missing references from previous literature. Please compare the reported BC levels with different devices under this work and the levels reported in previous studies in similar regions or of similar spatial characteristics.

Some references to previous studies in Helsinki have been added to section 3.2.1.

14. Section 3.2.4: Change the section title. I recommend removing this section and adding a detailed discussion explaining the variability in previous sections. If the authors want to keep this section, will this require some detailed analysis explaining how spatial variability is captured by different devices? If there are any differences in performances? Also, some discussion was required about how spatial variability can be captured by these devices. If the true spatial variability is higher/lower than the interdevice variability studied. Finally, some recommendations /comments would be helpful for the community, such as which device performed best in what context.

This section is removed and merged with section 3.2.1.

**15. After restructuring the result and discussion section - please update the conclusion accordingly. Avoid including conclusions that have not been discussed well in previous sections (For example, Line 436).**

The conclusions have been updated to correspond to the results and discussion provided in the manuscript.

---

## Referee Report (RR1)

Review of ar-2024-12: "The applicability and challenges of black carbon sensors in dense monitoring networks"

The paper elaborates on the applicability of BC sensors in dense monitoring networks. The figures are clear and easy to understand and the topic is of high interest especially in the frame of the upcoming EU regulations. I still have some minor comments listed below.

**General comments:**

To me, the word "dense" is a little misleading since the focus of the work is on BC small-scale sensors. I would remove the word dense. Because: What are the specific challenges with this BC sensors in the frame of a dense network over a stand-alone device included in a nation-wide network? Why is this emphasized?

Regarding the "Results and Discussion" section: For the intercomparison, to check whether or not the offset in the orthogonal fit is real, please elaborate whether or not a sufficient zero test was conducted within the intercomparison period. Also, please provide statistically backed estimates on the validity of the coefficients of the orthogonal regression. If yes and successful, why the calibration curve was not forced through zero?

Also, the text needs a revision in terms of interpunctuation and harmonization of how units are given (with or without space).

And last but not least, the paper would highly benefit from a more statistical analysis on the significance of the results (see below).

Please also set your work in context to other work like (Wu et al, 2023, https://doi.org/10.1016/j.jes.2023.05.044. who also checked the performance of protable BC monitors in field measurements

**Specific comments:**

p. 2 l. 50: space between number and unit. Please also harmonize along the text.

p. 2 l. 56: small-scale versions of? Word missing

p. 2 l. 58: In previous studies – add a comma there (check missing comma)

p. 3 l. 70: Please elaborate on the assumed performance of the MAAP. How you know, that this device can act as a reference?

p. 3 l. 79: An Aethalometer® is a device (and please use the registered trademark sign), not a method. Please rephrase.

p. 3 I. 84: "The measured variable by the instrument is the attenuation coefficient bATN( $\lambda$ ) [m-1] calculated from the measured attenuation" – if it's calculated its not measured.

p. 6 l. 155: According to Müller et al. (2011), the MAAP LED differs from the manufactures given wavelength and is 637 nm.

p. 7 Table 1: The given detection limit is given for a defined sampling period. Please elaborate. E.g., MA200 is "30 ng BC/m 3, 5 min timebase., 150 ml/min flow rate, SingleSpotTM" according to the technical specifications.

p. 12 l. 228: Please provide details on how the intercomparison was set up, e.g., whether or not the sensors sampled through the same inlet etc.

p. 22. L 368 and 369: "At all the locations, Mon and Tue had higher concentrations than Wed and Thu, and at most sites the highest concentrations were observed on Fri." Are these differences statistically significant – please add and/or comment within the text. This also expands to the later following analysis regarding the differences along sensors and weekdays.

p. 23 l. 387: I assume that the planetary boundary layer is thicker this time and the dilution of the emissions is enhanced. Please check.

p. 23 and p 24. Fig 12 and 13: The shown temperature clearly exceeds the operational temperature range given by the manufacturer. The question is whether or not the data is reliable.

Also, the 880 nm channel is near IR and hence sensitive to temperature changes. Please provide insights, if the bias was also visible in the other channels of the MA200. To me, a MAAP at 637 nm wavelength indicates that the 625 nm channel of the MA200/MA350 would also be suitable to determine the eBC mass concentration. Since the MAs provide multiple wavelengths, and the question of applicability of those sensors in a dense network is stated, also other channels should be considered. The effect of BrC should be negligible still.

Also, since the change rates of environment are of important I'd like to see also the change of the sample RH within the figure.

In a supplementary material I'd like to see the time series, showing that Observair sensors is significantly less affected by environmental changes.

p. 27 Table A1. I am missing the respective information on the MAAP at SMEAR III ground station.

---

## Referee Report (RR2)

The comparison of the performance of low-cost black carbon (BC) sensors, the correction algorithms and their impact on the measurements, as well as the sensitivity of these sensors to changes in relative humidity and temperature presented in this manuscript is valuable to the scientific community. Thus, I find this article suitable for publication, once the remaining points that need further attention are addressed.

- Although the language in the manuscript has improved, there are still instances where the writing lacks clarity. I suggest another round of language revision. For example, sentences like "Unfortunately, the Observair sensors are not being produced as of the end of 2023...", "The assumptions are that with 880 nm light source the absorption" or "However, still in the field study, several issues were observed..." need to be revised for better readability and precision.
- In Figure 2, the dates on the x-axis have not been corrected as suggested by Referee #1. There is still a shift in the dates between late May and early June. I recommend using consistent labeling, as with the other dates (every 2 days). Additionally, there is a language error: "Timeseries" should be corrected to "Time series."
- In Figures 12 and 13, there are errors in their descriptions. The units should be placed outside the parentheses, or the word "in" should be included within the parentheses.
- The subsections on methodology and results were not fully renamed to reflect the context, as suggested by both referees. Below are some examples for improved subsection titles:
  - 2.1. Measuring principle to obtain BC mass concentration with low-cost sensors
  - 2.3. Deployment of small BC sensors at the Kumpula Campus
  - 2.4. Description of the sampling site
  - $\circ~$  3.1. Intercomparison of BC sensors
  - 3.1.2. Adjusting differences between sensors for comparison
  - 3.2. Temporal and spatial variability during deployment
  - 3.2.1. BC levels during days of the week
  - 3.2.2. Diurnal variation in BC concentration
  - 3.2.3. Artifacts caused by sensor overheating
- The authors did not provide the strong justification in line 71 as suggested by Referee #2 on the use of the MAAP as the reference instrument.
- While presenting the calibrations results (F and orthogonal fit), a comparison with previous literature, as suggested for Referee #2 is missing. The authors report the results of the orthogonal fit (slope, correlation coefficients and intercept) in Table 4 but they do not include metrics from literature data with the type of calibration procedure adopted.
- The added paragraph discussing the results with literature on temporal and spatial variability has enhanced the revised version of the article, as suggested by the referee #2. However, the discussion could be further enriched by comparing these results in greater detail with findings from other studies conducted in urban areas across Europe using the same sensors.
- The discussion on spatial variability has improved; however, there are still questions raised by referee #2 that have not been yet fully addressed, which would be of interest to the reader. For instance: "How is spatial variability captured by different devices?" or "Are there differences in performance, and how can spatial variability be effectively captured by these devices?" Additionally, some recommendations or comments are missing, such as identifying which devices performed best in specific contexts.
- The authors have revised the conclusions in the manuscript; however, they need to update them by incorporating a few lines addressing the points raised here (previously asked by the referees). This would enhance the manuscript by providing valuable information and recommendations for the community. For example, identifying which device performed best for monitoring the spatial and temporal variability of BC mass concentrations under the specific

ambient conditions of the site (e.g., high relative humidity) would be particularly useful. Additionally, positioning the sampling site as having high, medium, or low BC levels in comparison to other urban sites using small BC sensors would further enrich the conclusions. This revision would avoid discussing aspects not explored in the manuscript, as noted by Referee #2, such as BC source apportionment, which refers to determining BC sources through spectral dependency or positive matrix factorization in the scientific community.

---

## Author Response (AR2)

**Point-to-point answers to reviewers**

The authors are grateful to the editor and reviewers who brought up important points and have spend their time in order to improve the manuscript. We are sorry that the review process had to be extended several times and we are grateful for the patience with this manuscript.

Below are point-to-point answers to each of the comments. The comments by the reviewers are marked bold, and the text copied from the revised manuscript is presented with cursive.

In addition updating the manuscript according to the comments, we updated Figs. 12 and 13, where we observed an error with reporting the value range of the temperature change variable. The change did not affect the shape of the curve. Therefore, fixing this error did not change the conclusions. Also, to improve clarity, we removed the "Both intercomparisons" -column from Table 4, as the values were not referred in the text or used in the data processing.

**Reviewer 1**

Review of ar-2024-12: "The applicability and challenges of black carbon sensors in dense monitoring networks"

The paper elaborates on the applicability of BC sensors in dense monitoring networks. The figures are clear and easy to understand and the topic is of high interest especially in the frame of the upcoming EU regulations. I still have some minor comments listed below.

**General comments**

To me, the word "dense" is a little misleading since the focus of the work is on BC smallscale sensors. I would remove the word dense. Because: What are the specific challenges with this BC sensors in the frame of a dense network over a stand-alone device included in a nation-wide network? Why is this emphasized?

We agree, removing word "dense" simplifies the title.

Regarding the "Results and Discussion" section: For the intercomparison, to check whether or not the offset in the orthogonal fit is real, please elaborate whether or not a sufficient zero test was conducted within the intercomparison period. Also, please provide statistically backed estimates on the validity of the coefficients of the orthogonal regression. If yes and successful, why the calibration curve was not forced through zero?

Unfortunately, no comprehensive zero tests were conducted during the intercomparison period or during the field deployment. Therefore, the intercept is included in the orthogonal regression. Standard errors of the orthogonal fit parameters were added in Table 4.

We added discussion of the studies that observed similar results of intercomparisons. To see these changes, please see the answer the Reviewer 2 comment starting with "The added paragraph discussing..." that is related to the same topic.

**Also, the text needs a revision in terms of interpunctuation and harmonization of how units are given (with or without space).**

We have modified the manuscript and paid focus on uniform notation for variables and units.

**And last but not least, the paper would highly benefit from a more statistical analysis on the significance of the results (see below).**

We understand this concern and we improved the statistical analysis, which has been added to Appendix A2 from the deployment phase. For the intercomparison periods error limits for fitting parameters have been added in Table 4, to show the accuracy of the fitting parameters.

References to statistical significance and to Appendix A2 have been added in the manuscript:

In Sect. 3.2.1: ... All locations had a statistically significant difference (Fig. A2.1), although the differences were not necessarily remarkable.

In Sect. 3.2.2: ... At all the locations, Mon and Tue had statistically significantly (Fig. A2.2) higher concentrations than Wed and Thu. The most stable locations were  $F_{ground}$  and SMEARIIIground, where there was no statistically significant difference between Mon, Tue, Fri, Sat and Sun. ....

**Please also set your work in context to other work like (Wu et al, 2023, https://doi.org/10.1016/j.jes.2023.05.044. who also checked the performance of protable BC monitors in field measurements**

Text that addresses other similar studies were added in various sections of the manuscript. Discussion referring to other studies was added in Sects. 3.1. and 3.2.1. To see these changes, please see the answer the Reviewer 2 comment starting with "The added paragraph discussing...".

Also, following text was added to Sect. 1 "Introduction":

Previous studies have reported a good correlation between BC sensors and reference-grade instruments, but with varying slopes and intercepts depending on location and sensor implicating the need for onsite calibration (Alas et al., 2020; Kuula et al., 2020; Chakraborty et al., 2023; Wu et al., 2024). In previous studies, a common application for these sensors has been personal BC exposure as a carry-on measurement device (Delgado-Saborit, 2012; Li et al., 2015), mobile measurements (Alas et al., 2019; Pikridas et al., 2019), and sensor networks (Caubel et al., 2019).

**Specific comments**

**p. 2 l. 50: space between number and unit. Please also harmonize along the text.**

We paid focus on uniform notation for variables and units throughout the manuscript.

**p. 2 l. 56: small-scale versions of? Word missing**

We added the missing word: ...small-scale versions of the filter-based instruments...

**p. 2 l. 58: In previous studies – add a comma there (check missing comma)**

Added a comma.

**p. 3 l. 70: Please elaborate on the assumed performance of the MAAP. How you know, that this device can act as a reference?**

MAAP is deployed at SMEAR III for long-term measurements. MAAP widely used in eBC monitoring and it is accepted instrument e.g., in the ACTRIS measurement infrastructure and it has been used as a reference instrument in previous studies (Alas et al., 2019; Luoma et al., 2021a). With the angular scattering measurements and applying the radiative transfer scheme in eBC retrieval, MAAP is less sensitive to aerosol particle scattering and is more independent method compared to, e.g., AE33.

The text was modified accordingly in Sect 2.3:

The reference instrument MAAP is also a filter-based absorption photometer, but it differs from the measurement principle presented in Sect. 2.1 by additionally measuring backscattering from the filter at two angles to improve the accuracy of the  $\sigma_{ap}$  and eBC. Additionally, the MAAP derives the  $\sigma_{ap}$  by applying a two-stream-approximation radiative transfer scheme (Petzold and Schönlinner, 2004). Therefore, it is somewhat more independent measurement method and is a good reference instrument for the eBC sensors. MAAP has been used as a reference instrument also in previous studies comparing filter-based instruments (Alas et al., 2019; Luoma et al., 2021a). The reported uncertainty and unit-to-unit variability of MAAP (at 16.67 l min-1 flow) are 12% and 3% (Petzold and Schönlinner, 2004; Müller et al., 2011). ...

**p. 3 l. 79: An Aethalometer® is a device (and please use the registered trademark sign), not a method. Please rephrase.**

Rephrased to: *Filter-based optical methods are widely used to measure BC concentration due to ease of operation and relatively low cost (Hansen et al., 1984).*

References to Aethalometer have been replaced with more generic language where appropriate. Trademark has been added when first referring to Aethalometer AE33 in Sect 2.1.

p. 3 l. 84: "The measured variable by the instrument is the attenuation coefficient  $bATN(\lambda)$  [m-1] calculated from the measured attenuation" – if it's calculated it's not measured.

This was fixed and rephased to: The attenuation coefficient  $b_{ATN}(\lambda)$  [m-1] is calculated from the measured light intensity and the operational parameters of the instrument as described in Eq. 2, ...

**p. 6 l. 155: According to Müller et al. (2011), the MAAP LED differs from the manufactures given wavelength and is 637 nm.**

This was corrected and the wavelength was changed to 637 nm and added a citation to Müller et al. (2011) in the text: *The instrument measures with only one wavelength at 637 nm (Müller et al., 2011) and applies MAC* =  $6.6 \text{ m}^2 \text{ g}^{-1}$  (at 637 nm).

p. 7 Table 1: The given detection limit is given for a defined sampling period. Please elaborate. E.g., MA200 is "30 ng BC/m 3, 5 min timebase., 150 ml/min flow rate, SingleSpotTM" according to the technical specifications.

We have updated the Table 1 according to the technical specifications.

**p. 12 l. 228: Please provide details on how the intercomparison was set up, e.g., whether or not the sensors sampled through the same inlet etc.**

This is described now in more detail in Sect. 2.4.: *The small-scale sensors were all measuring on the same sample line (different with MAAP) that did not have any inlet pre-impactor. The separate measurement line was set up through the SMEAR III station wall at a height of 3 m from the ground.*

p. 22. L 368 and 369: "At all the locations, Mon and Tue had higher concentrations than Wed and Thu, and at most sites the highest concentrations were observed on Fri." Are these differences statistically significant – please add and/or comment within the text. This also expands to the later following analysis regarding the differences along sensors and weekdays.

Added appendix A2 where statistical significance is given with the Wilcoxon signed-rank test (using scipy.stats.wilcoxon) between the locations and between the weekdays for every location.

Added and modified text in Sect. 3.2.2.

A daily breakdown can be seen in Fig. 10. There is somewhat surprising variation on day-today basis, as no notable differences were expected between weekdays. At all the locations, Mon and Tue had statistically significantly (Fig. A2.2) higher concentrations than Wed and Thu. The most stable locations were  $F_{ground}$  and SMEARIIIground, where there was no statistically significant difference between Mon, Tue, Fri, Sat and Sun. Therefore, weekend and weekdays did not seem to have a clear difference in the medians to each other, which is differing compared to other studies that observed lower eBC concentrations during weekends at traffic and at urban background sites in Helsinki (Helin et al., 2018; Luoma et al., 2021b). Also, Caubel et al. (2019) reported lower concentrations during weekends especially on traffic influenced cites. It is to be noted that the variance in concentrations was higher during Fri-Sun than during Mon–Thu and the highest peaks were measured during the weekend. The unexpected similarity between the weekdays and weekends might be due to a rather short period (74 d) for such an analysis and the time of the deployment period, which is a vacation season in Finland, when the anthropogenic activities are expected to depend less on the days of the week.

Also added text in Sect. 3.2.1:

All locations had a statistically significant difference (Fig. A2.1), although the differences were not necessarily remarkable.

**p. 23 l. 387: I assume that the planetary boundary layer is thicker this time and the dilution of the emissions is enhanced. Please check.**

This is now mentioned more clearly in the text in Sect. 3.2.3 where we modified the paragraph: The diurnal variation of eBC can be seen in Fig. 11, which shows a similar diurnal pattern at all the locations. The variation is affected by the local and regional anthropogenic activities and meteorological conditions. The eBC concentrations sharply rose during the morning due to increase in traffic. The highest concentrations were reached between 9-10 after which the concentrations decreased due to smaller traffic rates, increased dilution in the convective boundary layer due to higher mixing height and increased wind speeds (e.g., Fig. S2 in Luoma et al., 2021b). Another rise in concentration was observed late in the evening around 21–23. This increase was much less compared to the morning peak. The increased levels during the evenings are probably caused by accumulation of pollutants in a more stable atmosphere when the mixing height is lower and the wind speeds are also generally lower. Based on observations made in Helsinki, (Järvi et al., 2009) reported that out of the meteorological parameters, the wind speed and mixing height had the greatest effect on eBC concentrations. Also, local wood combustion emissions, for example, evening activities at the close by community garden, can increase the eBC levels. Similar diurnal patterns with a peak in the morning and evening have been observed by previous studies during the warm period at traffic and urban background sites (Sahu et al., 2011; Backman et al., 2012; Caubel et al., 2019; Luoma et al., 2021b).

p. 23 and p 24. Fig 12 and 13: The shown temperature clearly exceeds the operational temperature range given by the manufacturer. The question is whether or not the data is reliable. Also, the 880 nm channel is near IR and hence sensitive to temperature changes. Please provide insights, if the bias was also visible in the other channels of the MA200. To me, a MAAP at 637 nm wavelength indicates that the 625 nm channel of the MA200/MA350 would also be suitable to determine the eBC mass concentration. Since the MAs provide multiple wavelengths, and the question of applicability of those sensors in a dense network is stated, also other channels should be considered. The effect of BrC should be negligible still. Also, since the change rates of environment are of important I'd like to see the time series, showing that Observair sensors is significantly less affected by environmental changes.

Anomalous activity is observed while below 40 °C between 6.00-8.00 (Fig. R2), which is within the operational temperature and should be considered reliable. We added a mention of this to Sect 3.2.4: According to the instrument manual, the operating temperature for the instrument is 0-40 °C. In our deployment, the T increases even to 52 °C, which is above the operating temperature and could explain the behaviour. However, the anomalous activity is observed even below the 40 °C.

We studied the observed behaviour also on other wavelengths, which are presented below in Fig. R1. The bias is visible on all wavelengths, but it is less pronounced at lower wavelengths. Other wavelengths were not used in the analysis to keep the scope of the study on eBC and to be comparable to the other sensor types. We added a mention of this to Sect 2.3:

... The other wavelengths of the MA-sensors can be used to differentiate between BrC and BC and the possible sources of these particles. In this study only the 880 nm wavelength was utilized to conform to the other sensors. ...

To see also the variation in RH, we have now added the RH change rate to Figs. 12 and 13.

In Fig. R2 the Observairs seem more stable than the individual spots of the MA350 between 6.00-10.00 although the *T* gradient is less. The main problem arises from that the instability of the individual spots is amplified by the dualspot correction. This is relevant if the dualspot corrected data is considered the default output of the instrument. The spiking in Observair sensors around 11 is due to handling of the instruments (just inspecting that the measurement is running via opening the deployment box). With the handling it could be argued that the Observair sensors too are susceptible to sharp changes in *T*. This enforces the requirement of robust deployment with *T* and *RH* controlled deployment boxes and that the algorithmic solutions are suitable for lower temperature gradients but do also have a limit (which is not determined in this study).

The conclusions are updated to emphasize that the user needs to take into account the changing environmental parameters more carefully. See the answer the Reviewer 2 comment starting with "The added paragraph discussing...".

Figure R1. MA350 all wavelengths.